# Structural and functional analyses of hepatitis B virus X protein BH3-like domain and Bcl-xL interaction

Tian-Ying Zhang [1,2,7], Hong-Ying Chen[3,7], Jia-Li Cao[1,2,7], Hua-Long Xiong[1,2], Xiao-Bing Mo[3], Tian-Liang Li[4], Xiao-Zhen Kang[1,2], Jing-Hua Zhao[1,2], Bo Yin[3], Xiang Zhao[5], Cheng-Hao Huang[1,2], Quan Yuan [1,2], Ding Xue[4,5], Ning-Shao Xia [1,2] & Y. Adam Yuan [3,6]

Hepatitis B virus (HBV) X protein, HBx, interacts with anti-apoptotic Bcl-2 and Bcl-xL proteins through its BH3-like motif to promote HBV replication and cytotoxicity. Here we report the crystal structure of HBx BH3-like motif in complex with Bcl-xL where the BH3-like motif adopts a short α-helix to snuggle into a hydrophobic pocket in Bcl-xL via its noncanonical Trp120 residue and conserved Leu123 residue. This binding pocket is ~2 Å away from the canonical BH3-only binding pocket in structures of Bcl-xL with proapoptotic BH3-only proteins. Mutations altering Trp120 and Leu123 in HBx impair its binding to Bcl-xL in vitro and HBV replication in vivo, confirming the importance of this motif to HBV. A HBx BH3-like peptide, HBx-aa113-135, restores HBV replication from a HBx-null HBV replicon, while a shorter peptide, HBx-aa118-127, inhibits HBV replication. These results provide crucial structural and functional insights into drug designs for inhibiting HBV replication and treating HBV patients.

[1] State Key Laboratory of Molecular Vaccinology and Molecular Diagnostics, School of Public Health and School of Life Sciences, Xiamen University, 361102 Xiamen, China. [2] National Institute of Diagnostics and Vaccine Development in Infectious Diseases, School of Public Health and School of Life Sciences, Xiamen University, 361102 Xiamen, China. [3] National University of Singapore (Suzhou) Research Institute, 377 Lin Quan Street, Suzhou Industrial Park, 215123 Jiangsu, China. [4] Department of Molecular, Cellular and Developmental Biology, University of Colorado, Boulder, CO 80309, USA. [5] School of Life Sciences and Collaborative Innovation Center for Diagnosis and Treatment of Infectious Diseases, Tsinghua University, 100084 Beijing, China. [6] Department of Biological Sciences and Centre for Bioimaging Sciences, National University of Singapore, 14 Science Drive 4, Singapore 117543, Singapore. [7] These authors contributed equally: Tian-Ying Zhang, Hong-Ying Chen, Jia-Li Cao. Correspondence and requests for materials should be addressed to Q.Y. (email: yuanquan@xmu.edu.cn) or to D.X. (email: ding.xue@colorado.edu) or to N.-S.X. (email: nsxia@xmu.edu.cn)

Hepatitis B virus (HBV) is a small DNA virus with a genome size around 3.2 kb. HBV infection stimulates the development of hepatitis and hepatocellular carcinoma (HCC), which is a leading cause of liver cancer worldwide[1,2]. The HBV genome encodes four viral gene products, one of which, the HBV X protein (HBx), plays an important role in HBV life cycle through interaction with a number of host proteins[3–8]. For example, HBx has been shown to interact with damage-specific DNA-binding protein 1 (DDB1), which could redirect the DDB1-contaning E3 ubiquitin ligase to target the structural maintenance of chromosome 5/6 complex (Smc5/6) for degradation, unleashing the transcriptional repression by Smc5/6 to increase HBV viral gene expression[6,9]. Fragments of HBV DNA are integrated into numerous sites of the host genome and the HBx gene is often over-expressed in the livers and tumors of HBV chronic carriers[10,11]. Hence, HBx actively participates in the development of chronic liver disease and HCC. Recently, HBx is found to directly target anti-apoptotic proteins Bcl-2 and Bcl-xL through a Bcl-2 homology region 3 (BH3)-like motif to induce the increase of cytosolic calcium, which is required for HBV viral replication, and also induces cytotoxic effects through apoptosis and necrosis, leading to HBV pathogenesis[7,12]. Although the structure of the HBx-BH3-like peptide/Bcl-2 complex was reported recently[8], the structural basis of the interaction between the HBx-BH3-like motif and the Bcl-xL protein and the impact of this interaction to the life cycle of HBV have not been determined.

In this study, we show that the HBx-BH3-like motif, which is distantly related to the canonical BH3-only motif[7,12], forms an atypical short amphipathic α-helix that snuggles into a distinct hydrophobic pocket in Bcl-xL. Unlike other published structures of Bcl-xL in complex with BH3-only-containing proteins, the HBx-BH3-like motif binds to a hydrophobic pocket ~2 Å away from the canonical pocket in Bcl-xL that mediates binding to other BH3-only domains. Two residues, Trp120 and Leu123, in the HBx-BH3-like motif closely interact with residues in the Bcl-xL binding pocket via hydrophobic interactions. Supporting this structural observation, a peptide comprising the HBx-BH3-like motif (HBx-BH3-aa113–135), but not an equivalent peptide harboring the W120A/L123A double mutations, is capable of pulling down the endogenous Bcl-xL protein from HepG2 cells. More importantly, this HBx-BH3-like peptide, but not an equivalent peptide harboring the W120A/L123A double mutations, is able to restore HBV replication and transcription in HepG2 cells transfected by a replication-defective HBx-null HBV replicon. Notably, a shorter HBx-aa118–127 peptide, but not two potent BH3-only mimetics (ABT-263 and ABT-737), significantly inhibits HBV replication and HBV S-antigen (HBsAg) expression in HepG2 cells and in infection-competent HepG2-NTCP cells at a dose–response manner, indicating that HBx-aa118–127 can serve as a drug lead for HBV treatment. Moreover, structural comparisons of the Bcl-xL/HBx-BH3-like complex with the Bcl-xL/ABT-263 and Bcl-xL/ABT-737 complexes reveal a distinct and novel hydrophobic pocket in Bcl-xL that is critical for binding to the Trp120 side chain of the HBx-BH3-like motif and that may account for the failure of canonical BH3-only mimetics, such as ABT-263 and ABT-737, in suppressing HBV replication. Hence, our structural and functional analyses of the HBx/Bcl-xL interaction not only identify key residues and the structural basis that mediate binding of HBx to Bcl-xL and a new and distinct binding pocket in Bcl-xL for the HBx-BH3-like motif but also point out the need to employ a new strategy to screen for unique HBx-BH3-like mimetics that can inhibit HBV replication and treat HBV-related liver disease.

## Results

**The overall structure of the Bcl-xL/HBx-BH3-like complex.** A series of Bcl-xL proteins with different lengths were expressed and purified for co-crystallization with HBx peptides containing the BH3-like motif. The co-crystallization process of most Bcl-xL proteins with the HBx-BH3-like peptide yielded only Bcl-xL crystals without the HBx peptide at medium resolutions. The co-crystals of Bcl-xLΔTM (named Bcl-xL thereafter), in which the flexible loop (residues 27–82) and the transmembrane domain (residues 197–233) are deleted, in complex with a HBx peptide (residues 113–135) yielded a complex that was diffracted to 2.15 Å resolution (PDB ID 5B1Z). There are two Bcl-xL molecules and two HBx peptides in one asymmetric unit. Residues 1–26 and 83–196 of Bcl-xL and residues 118–135 of the HBx peptide are well refined. Upon binding with the HBx peptide, the second half of the α3-helix (residues 107–113) of Bcl-xL shifts ~5 Å away from the first half of the α4-helix (residues 118–124) and, together with the loop connecting the α3 and α4 helices (residues 114–117), form a small binding cleft to accommodate the HBx-BH3-like motif (Fig. 1a, Supplementary Fig. 1). As expected, the structure of the bound HBx-BH3-like motif (residues 118–127) adopts an amphipathic α-helix in the complex (Fig. 1a, Supplementary Fig. 1). Notably, the α-helix formed by the HBx-BH3-like motif contains only two helical turns, which is shorter than those of all published structures of the BH3-only domain proteins in complex with Bcl-xL (Fig. 1b, Supplementary Fig. 2a–f) (PDB ID 1PQ1)[13–15]. The C-terminal residues of the bound HBx-BH3-like peptide form a random coil fitting into the narrow groove at the top of the short α-helix-bound region (Fig. 1d). Hence, the binding pocket in Bcl-xL for the HBx-BH3-like motif is much smaller than that observed in Bcl-xL in complex with the Bim-BH3-only motif (Fig. 1d, e)[13]. Compared with the long α-helix formed by the Bim-BH3-only domain bound to Bcl-xL, less significant conformational change of Bcl-xL is observed upon binding of the HBx-BH3-like motif (Fig. 1d–f). Moreover, the short α-helix of the HBx-BH3-like motif slides 2 Å away from the canonical BH3-only binding pocket of Bcl-xL (Supplementary Fig. 2b). Therefore, the HBx-BH3-like amphipathic α-helix interacts with Bcl-xL through different residues in Bcl-xL (Fig. 1a–c).

**Structural roles of the HBx-BH3-like motif in Bcl-xL binding.** The interaction between Bcl-xL and the HBx-BH3-like motif is primarily mediated by hydrophobic interactions. The side chain of Trp120 in the HBx-BH3-like motif (corresponding to the Leu residue in the canonical BH3 motif) is surrounded by the hydrophobic side chains of Val126 and Phe146 in Bcl-xL at one side and Phe105, Leu108, and Leu130 at the other side (Fig. 1a, c, d). The side chain of Leu123 in the HBx-BH3-like motif packs intimately into a hydrophobic pocket formed by the side chains of Ala104 and Phe105 in Bcl-xL (Fig. 1a, c). The side chain of Ile127 in the HBx-BH3-like motif fits well into the hydrophobic pocket formed by Phe97, Tyr101, and Ala142 in Bcl-xL (Fig. 1a). Notably, three residues, Trp120, Leu123, and Ile127, are located at one side of the short α-helix with their side chains pointing towards the hydrophobic pocket formed by Bcl-xL (Fig. 1a, c). Moreover, a hydrogen bond is formed between the main chain of Leu194 in Bcl-xL and the side chain of Asp130 in the HBx-BH3-like motif. Hence, the tight hydrophobic interactions contributed by the side chains from the short α-helix in the HBx-BH3-like motif, supplemented by a hydrogen bond contributed by the Asp130 residue at the C-terminal end of the α helix, provide tight binding between Bcl-xL and the HBx-BH3-like motif. Besides the aforementioned interactions, there is no other interaction between Bcl-xL and the HBx-BH3-like motif. The conserved Asp

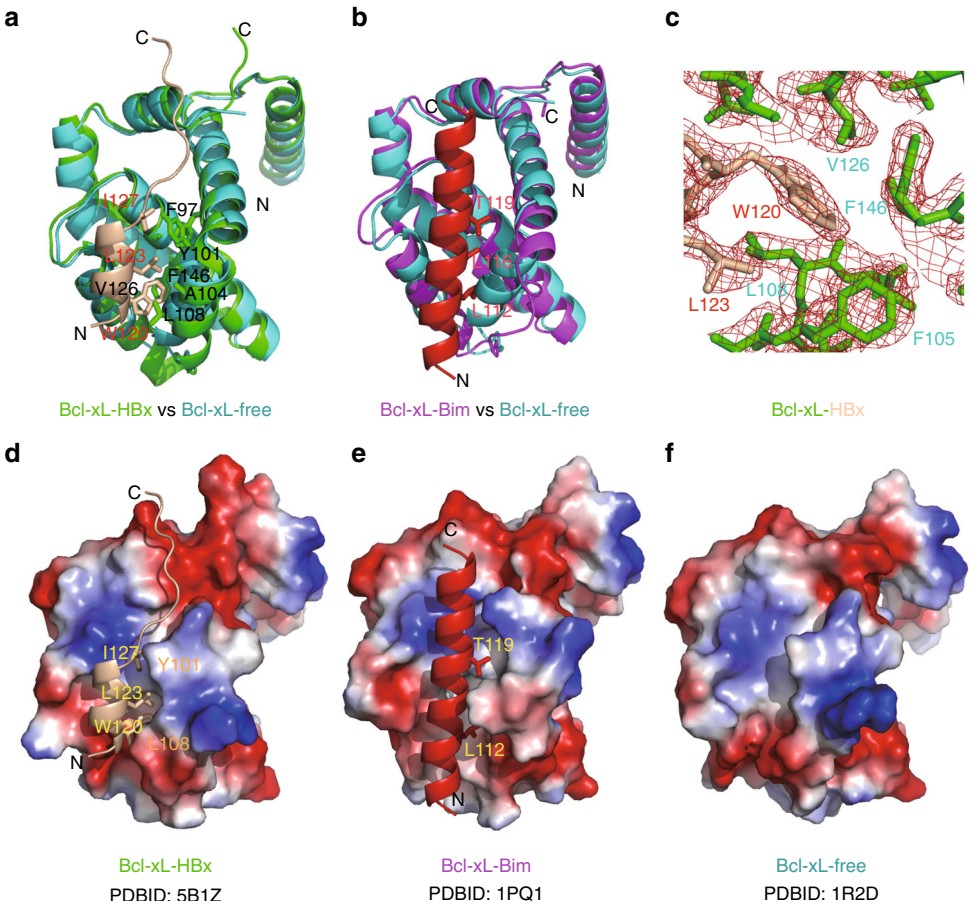

**Fig. 1** The crystal structure of the Bcl-xL/HBx-aa113–135 complex. **a** Cartoon representation of structural comparison of Bcl-xL/HBx-BH3 motif with Bcl-xL at free form (PDB ID 5B1Z). The structures of Bcl-xL/HBx-BH3 motif and Bcl-xL at free form are colored in green and cyan, respectively. The bound HBx-BH3 peptide is shown in carton mode and colored in wheat. The key residues involved in tight interactions are highlighted in stick mode. **b** Cartoon representation of structural comparison of Bcl-xL/Bim-BH3 motif (PDB ID 1PQ1[13]) with Bcl-xL at free form (PDB ID 1R2D[46]). The structure of Bcl-xL/Bim-BH3 motif is colored in magenta. The bound Bim-BH3 peptide is shown in carton mode and colored in red. The corresponding key residues involved in tight interactions are highlighted in stick mode. **c** Electrostatic potential surface view of Bcl-xL/HBx-BH3 motif complex. The bound HBx-BH3 motif forms a short, two-helical-turn α-helix snuggling into a small hydrophobic pocket, which is partially preformed. The key residues involved in hydrophobic interactions are indicated. **d** Electron densities around the HBx W120 residue are shown in red mesh contours. **e** Electrostatic potential surface view of Bcl-xL/Bim-BH3 motif complex. The bound Bim-BH3 motif forms a continuous long α-helix fitting into the big groove formed by Bcl-xL upon Bim binding. The corresponding key residues involved in interactions are indicated. **f** Electrostatic potential surface view of Bcl-xL at free form. A preformed small groove/pocket at the bottom of the surface is observed. **d–f** were made by pymol using the default parameters provided. The map was calculated using Fo-Fc coefficients and phases from the refined structure, but with the residues omitted from the Fc calculation. The map is contoured at 3σ. The side chains of HBx-BH3 and Bcl-xL are shown in stick model. HBx and Bcl-xL molecules are colored in wheat and green, respectively

residue in the signature Gly-Asp motif in the canonical BH3 domain is replaced by a Glu residue in the HBx-BH3-like motif, which does not appear to interact with any residue in Bcl-xL. These findings indicate that the HBx-BH3-like motif interacts with Bcl-xL through a distinct binding pocket and drastically different side-chain interactions.

**HBx interacts with Bcl-xL through the BH3-like domain.** To investigate if the HBx-aa113–135 peptide used in the structural study is sufficient to mediate binding of HBx with the endogenous Bcl-xL protein in human hepatic HepG2 cells, we performed the co-precipitation (co-IP) assays. The wild-type (WT) HBx-aa113–135 peptide and the corresponding mutant HBx-aa113–135(W120A/L123A) peptide with a C-terminal lysine-biotinylation tag were synthesized and used to precipitate the Bcl-xL protein from the HepG2 cell lysate. As expected, Bcl-xL was co-precipitated from the HepG2 cell lysate by the WT HBx-

aa113–135 peptide using streptavidin-conjugated magnetic beads. By contrast, much less Bcl-xL was co-precipitated by the HBx-aa113–135(W120A/L123A) peptide (Fig. 2a).

To examine if the Trp120 and Leu123 residues in the BH3-like motif of HBx are critical for mediating the interaction between HBx and the endogenous Bcl-xL protein, an HBx expression vector, pTT22-mCherry-HBx, or its mutant version, pTT22-mCherry-HBx(W120A/L123A), was transfected into HepG2 cells and the interaction between HBx and Bcl-xL was examined by co-IP assays using an anti-HBx antibody, 16F9. Consistent with the results from the pull-down assays using the biotinylated HBx-aa113–135 peptides, the W120A/L123A double mutations significantly weakened the interaction between Bcl-xL and the full-length HBx protein in HepG2 cells (Fig. 2b). Furthermore, isothermal titration calorimetry (ITC) assays were used to measure the binding affinities between the HBx-aa113–135 peptide and the Bcl-xL protein and between the HBx-aa113–135(W120A/L123A) mutant peptide and the Bcl-xL

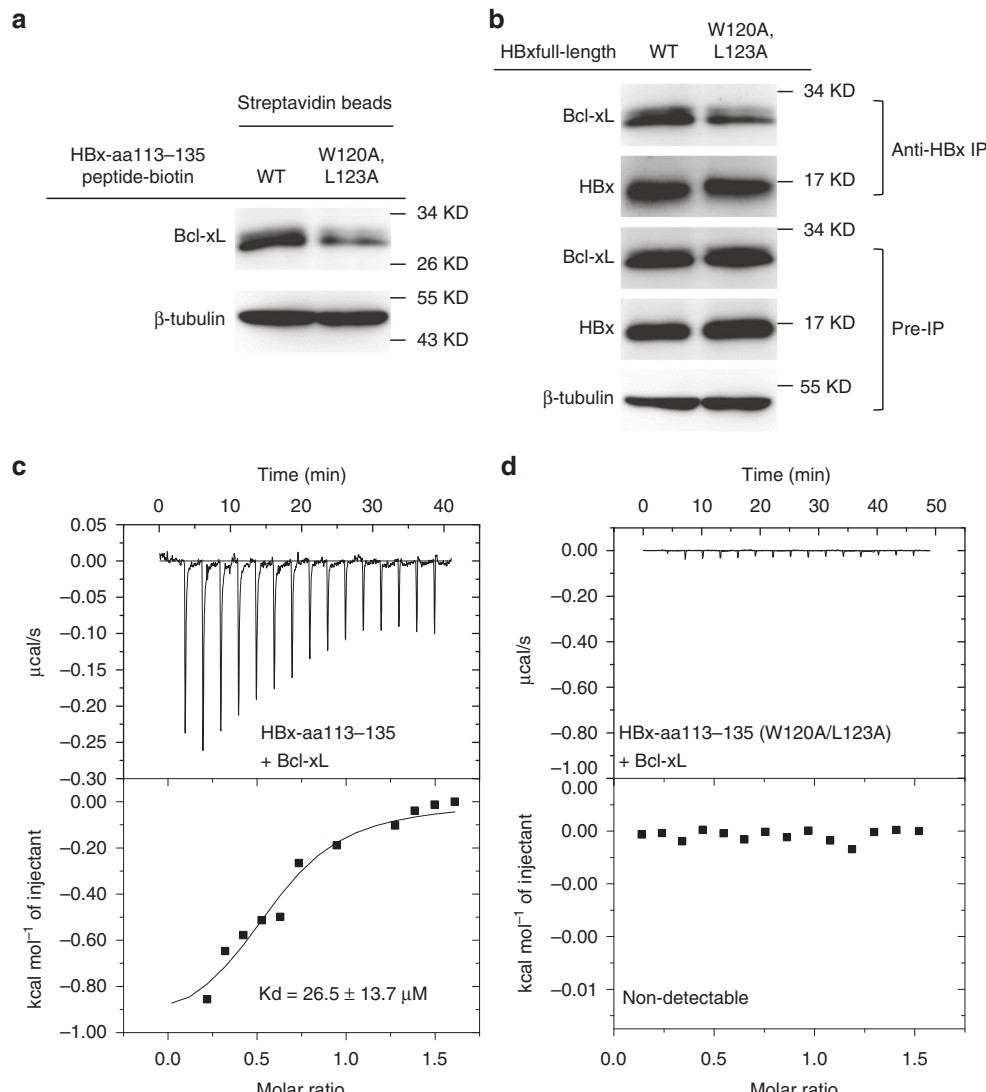

**Fig. 2** HBx binds endogenous Bcl-xL through the BH3-like domain. **a** HBx-aa113–135 interacts with Bcl-xL in HepG2 cell lysate. Co-precipitation experiments were performed in the HepG2 cell lysate using biotinylated HBx-aa113–135 peptides (wild type (WT)) and corresponding mutant peptides (W120A/L123A), respectively. The results were analyzed by immunoblotting (IB) using anti-Bcl-xL antibody. β-Tubulin was measured and analyzed as an input control. **b** Full-length HBx interacts with Bcl-xL in HepG2 cells. Co-immunoprecipitation (co-IP) experiments were performed in HepG2 cells transfected with pTT22m-HBx-WT or pTT22m-HBx(W120A/L123A). The cell lysate was precipitated by an anti-HBx antibody and analyzed by immunoblotting using an anti-Bcl-xL antibody. **c** Isothermal titration calorimetry (ITC) result shows the dissociation constant between HBx113-135 and Bcl-xL is 26.5 ± 13.7 μM. **d** The interaction between HBx-aa113–135(W120A/L123A) and Bcl-xL is non-detectable by ITC. Source data are provided as a Source Data file

protein, respectively. The data showed that the dissociation constant between HBx-aa113–135 and Bcl-xL was 26.5 ± 13.7 μM (Fig. 2c), whereas the interaction between HBx-aa113–135 (W120A/L123A) and Bcl-xL was not detectable (Fig. 2d). These results confirm that Trp120 and Leu123 indeed are two residues in the HBx-BH3-like motif crucial for the interaction between Bcl-xL and HBx.

**The HBx-BH3-like peptide promotes HBV production**. We investigated if the HBx-aa113–135 peptide alone is capable of promoting HBV viral gene expression and DNA replication in HepG2 cells. We first set up an efficient peptide delivery method by embedding the target peptide in nanoparticles formed by a peptide carrier, PEP-1 (Fig. 3a)[16]. An HBx-null HBV viral replicon (pHBV1.3-Xnull), in which an early stop codon is introduced to the HBx coding region in a 130% replication-competent HBV genome, was transfected into HepG2 cells for 6

h, which were then incubated with control peptide/PEP-1, HBx-aa113–135/PEP-1, or HBx-aa113–135(W120A/L123A)/PEP-1 nanoparticles for an additional 48 h. The expression levels of HBsAg, a key barometer of HBV infection, and HBV e-antigen (HBeAg) in the supernatant of transfected HepG2 cells and the expression levels of HBV core antigen (HBcAg) in transfected HepG2 cells were quantified (see Methods). As expected, the levels of the HBV viral protein expression and HBV viral replication in HepG2 cells transfected with pHBV1.3-Xnull and treated with control peptide/PEP-1 were low (Fig. 3a–e), because the pHBV1.3-Xnull replicon largely failed to replicate in HepG2 cells due to the absence of the HBx protein, which is critical for HBV viral replication[17–19]. Remarkably, incubation of HBx-aa113–135/PEP-1 nanoparticles with HepG2 cells transfected with pHBV1.3-Xnull significantly increased both HBV viral protein production and HBV DNA replication in transfected HepG2 cells (Fig. 3b–e), whereas incubation with HBx-aa113–135

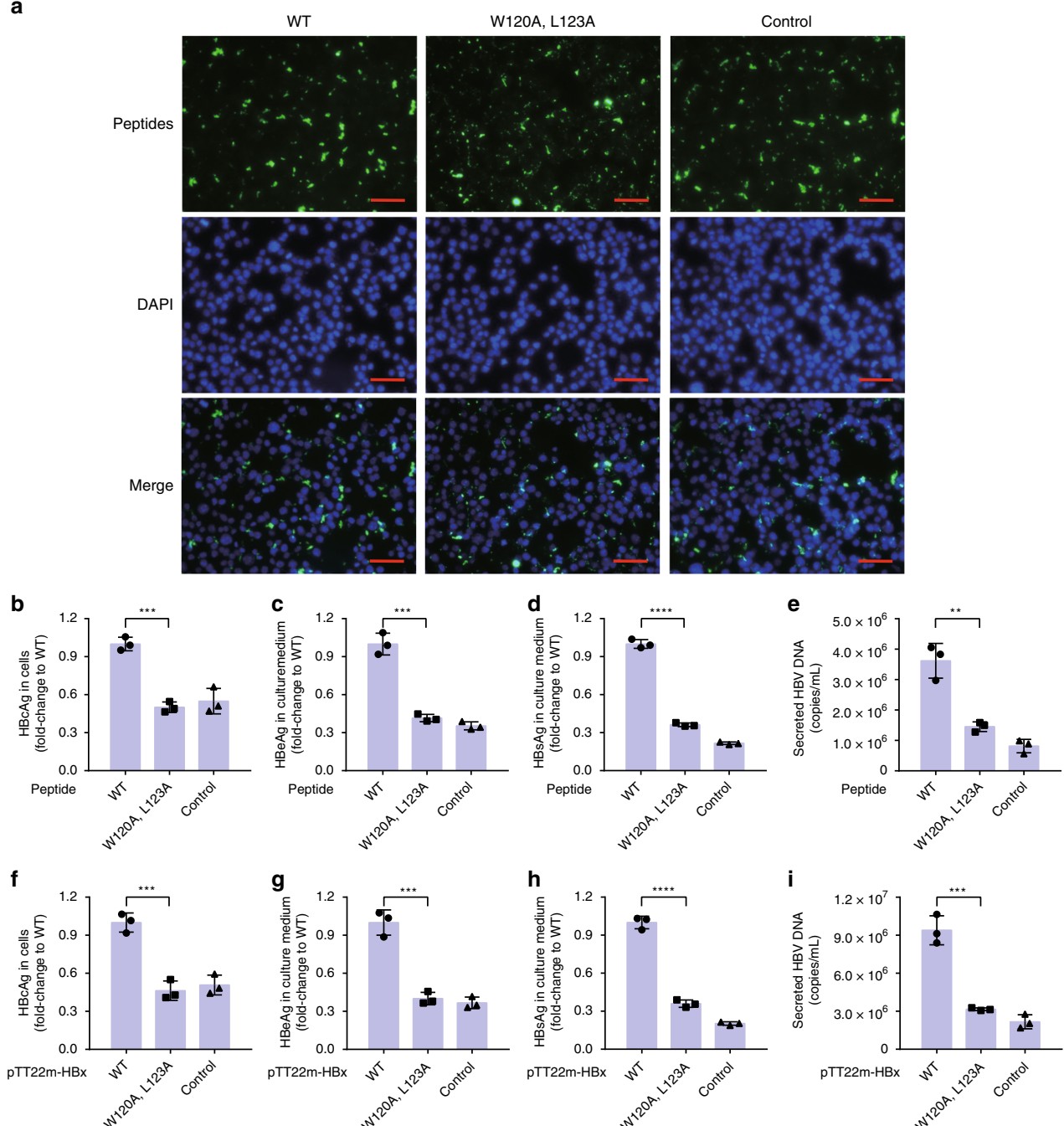

**Fig. 3** The HBx-BH3-like domain is critical for HBV production in HepG2 cells. **a–e** C-terminal biotinylated HBx-aa113–135 peptides (both wild-type (WT) and the W120A/L123A double mutant) were delivered into HepG2 cells transfected with pHBV1.3-Xnull using PEP-1 nanoparticles (Methods) and analyzed by streptavidin-FITC (fluorescein isothiocyanate) for transfection efficiency (**a**). Scale bar is 50 μm. An unrelated 23-aa peptide was used as a control (Control). The intracellular expression levels of HBV core antigen (HBcAg (**b**) and the levels of HBV e-antigen (HBeAg) (**c**), HBV S-antigen (HBsAg) (**d**), and HBV DNA (**e**) in the culture medium collected from HepG2 cells were measured (Methods). The results are shown as fold change to the WT HBx-aa113–135 peptide (WT). **f–i**) HepG2 cells were co-transfected with pHBV1.3-Xnull and pTT22m-HBx-WT (WT), pHBV1.3-Xnull and pTT22m-HBx (W120A/L123A), or pHBV1.3-Xnull and the pTT22m vector control (Control). Intracellular expression levels of HBcAg (**f**) and the levels of HBeAg (**g**), HBsAg (**h**), and HBV DNA (**i**) in the culture medium from HepG2 cells were measured similarly. The results are shown as fold change to the combination of pHBV1.3-Xnull and pTT22m-HBx-WT (WT). Data shown represent mean ± SD. Significance is indicated on the top. **P < 0.01, ***P < 0.001, ****P < 0.0001; two-tailed unpaired t tests (n = 3 biologically independent samples). Source data are provided as a Source Data file

(W120A/L123A)/PEP-1 nanoparticles failed to do so. These results indicate that the HBx-aa113–135 peptide alone is sufficient to restore HBV replication and viral protein expression in HepG2 cells from replication-defective HBV replicons lacking a functional HBx.

To further validate this finding, the pHBV1.3-Xnull replicon and the pTT22-mCherry-HBx plasmid expressing full-length HBx or the HBx (W120A/L123A) mutant were co-transfected into HepG2 cells. The expression levels of HBsAg and HBeAg and HBV viral DNA in the supernatant, as well as the expression

level of HBcAg in transfected HepG2 cells were quantified. Consistent with results from the peptide delivery assays, the levels of HBV viral protein production and viral DNA replication in HepG2 cells co-transfected with pHBV1.3-Xnull and pTT22-mCherry-HBx were significantly higher than those co-transfected with pHBV1.3-Xnull and pTT22-mCherry-HBx(W120A/L123A) or the pTT22-mCherry vector control (Fig. 3f–i). These results together demonstrate for the first time that the 23-residue HBx-BH3-like peptide alone is capable of promoting both HBV viral gene expression and DNA replication.

**HBx Trp120 and Leu123 residues are critical for HBV production**. To investigate the importance of the interface residues, Trp120 and Leu123, in the HBx-aa113–135/Bcl-xL complex to HBV viral replication, we examined HBV replication in HepG2 cells transfected with the pHBV1.3 replicon without (WT) or with W120A/L123A mutations (pHBV1.3-WL/AA) by Southern blot analysis. Compared with HepG2 cells transfected with the WT pHBV1.3 replicon, HBV replication was greatly reduced in cells transfected with pHBV1.3-WL/AA or pHBV1.3-Xnull (Fig. 4a). Northern blot analysis revealed that the levels of the HBV transcription also markedly decreased in these HepG2 cells (Fig. 4b), which correlated well with the levels of HBV viral DNA replication (Fig. 4a).

We also examined the importance of these interface residues, Trp120 and Leu123, to HBV viral replication in a mouse HBV model[20]. pHBV1.3, pHBV1.3-Xnull, and pHBV1.3-WL/AA replicons were hydrodynamically injected into the C57BL/6 mice. The levels of HBsAg and HBeAg in the serum and the levels of intrahepatic HBcAg and HBsAg were measured by chemiluminescent enzyme immunoassay. The intrahepatic expression levels of HBsAg and HBcAg were also analyzed by immunohistochemistry assays. Compared with mice injected with the WT pHBV1.3 replicon, the serum and intrahepatic expression levels of HBV viral proteins in mice injected with pHBV1.3-WL/AA were significantly lower, which were comparable to those observed in mice injected with pHBV1.3-Xnull (Fig. 4c–h). These results demonstrate that the HBx-BH3-like motif and its two interface residues, Trp120 and Leu123, are critical for multiple aspects of the HBV life cycle.

We last validated the importance of the interface residues, Trp120 and Leu123, in supporting HBV infection in HepG2-NTCP cells, which is a more physiological model for HBV replication and viral protein expression[21,22]. Recombinant HBV particles, including HBV-WT, HBV-WL/AA, and HBV-Xnull, were produced by transfection of Huh7 cells with pHBV1.3, pHBV1.3-Xnull, and pHBV1.3-WL/AA replicons and their viral titers were normalized to be equal for infection of HepG2-NTCP cells. Consistent with the in vivo data from the hydrodynamically injected HBV mouse model, the levels of HBsAg and HBeAg in the culture medium of the HBV-WL/AA-infected HepG2-NTCP cells were much lower than those from HBV-WT-infected HepG2-NTCP cells, while viral antigen levels were barely detectable in HBV-Xnull-infected HepG2-NTCP cells (Fig. 4i, j). These results together confirm that the HBx interface residues with Bcl-xL, Trp120, and Leu123 and the interaction of HBx with Bcl-xL are critical for the HBV life cycle.

**Structures of Bcl-xL/BH3-like peptide and Bcl-xL/BH3 mimetics**. Recently, the structure of Bcl-2/HBx-BH3-like complex was reported[8]. In this structure, the HBx-BH3-like peptide (HBx-aa110–135) was used for co-crystallization with a modified Bcl-2 protein (PDB ID 5FCG[8]). Surprisingly, although the structures of Bcl-2 and Bcl-xL in these two complexes, Bcl-2/HBx-aa110–135 and Bcl-xL/HBx-aa113–135, appear similar (Z-score 16.7, Cα 120,

root-mean-square deviation (r.m.s.d.) 2.7 Å), the HBx-BH3-like domains in these two complexes adopt significantly different structures (Supplementary Fig. 3a). In the Bcl-2/HBx-aa110–135 structure[8], HBx-aa110–135 (residues 113–135 are observed and built) adopts an amphipathic α-helix fold that has the extended conformation at its C-terminal portion and makes fewer interactions with Bcl-2. By contrast, in the Bcl-xL/HBx-aa113–135 structure, HBx-aa113–135 (residues 118–135 are observed and built) displays a short α-helix with two helical turns. Compared with the Bcl-xL/HBx-aa113–135 structure, the BH3-like motif in the Bcl-2/HBx-aa110–135 structure is shifted more than 8 Å towards the N-terminus (Supplementary Fig. 3a). Hence, the side chains of the key residues (Trp120, Leu123, and Ile127) in the HBx-BH3-like motif from these two structures interact with different sets of residues in Bcl-2 and Bcl-xL, respectively (Supplementary Fig. 3a). Specifically, in the Bcl-2/HBx-aa110–135 structure[8], the side chain of Trp120 is located at the shallow hydrophobic groove formed by Gly89 and Cys90, which is ~8 Å away from the hydrophobic pocket that the Trp120 side chain binds to in the Bcl-xL/HBx-aa113–135 structure (Supplementary Fig. 3a). Based on the Bcl-2/HBx-aa110–135 structure, a slightly different BH3 motif, LGEEIRLKV, was proposed[8], which does not include the Trp120 and Leu123 residues that are critical for HBV replication and viral gene expression in our multiple HBV life cycle assays (Figs. 3 and 4).

To further analyze the novel hydrophobic pocket in Bcl-xL bound by the HBx Trp120 side chain, the published structures of Bcl-xL/ABT-263 (PDB ID 4QNQ) and Bcl-xL/ABT-737 (PDB ID 2YXJ[23]) complexes were superimposed with the Bcl-xL/HBx-aa113–135 structure (Fig. 5e, f). These three structures can be largely superimposed (Z-score 19.8–19.9, Cα 137–140, r.m.s.d. 0.6–0.9 Å). The buried surface area in the Bcl-xL/HBx-aa113–135 complex is 2014.76 Å2 per molecule, which is slightly larger than those of the Bcl-xL/ABT-263 and Bcl-xL/ABT-737 complexes (Supplementary Table 1), suggesting that the Bcl-xL/HBx-aa113–135 structure is physiologically relevant. Notably, the Trp120 side chain of HBx inserts into the hydrophobic pocket formed by Phe105, Leu108, Val126, and Phe146 of Bcl-xL (Figs. 1c, 5a, and Supplementary Fig. 3b), whereas ABT molecules bind to two shallow grooves formed by the side chains of Phe97/Tyr195 and Phe97/Phe146 in Bcl-xL (Fig. 5b, c and Supplementary Fig. 3b). There is ~10 Å distance between the center of ABT molecule and the side chain of Trp120 (Supplementary Fig. 3b). Hence, the hydrophobic pocket occupied by the critical HBx Trp120 side chain could be a novel druggable site unique for blocking HBV replication and treating HBV. Moreover, given the structural feature of the HBx-aa113–135/Bcl-xL complex and the importance of Trp120 in HBx binding to Bcl-xL and HBV viral replication and expression, we suggest that the "WEELGEEI" motif should be the right BH3-like domain within HBx, as it was first defined[7,12].

**HBx-aa118–127 peptide inhibits HBV replication**. We took advantage of the structural information of the HBx-aa113–135/Bcl-xL complex to develop potential HBV inhibitors. HepG2.2.15 cells, which actively replicate HBV[24], were treated with HBx-aa113-135, a peptide that can promote HBV replication (Fig. 3), and HBx-aa118-127, a shorter peptide that harbors the key residues for binding to Bcl-xL and may become a dominant-negative inhibitor. BH3-only mimetics, ABT-236 and ABT-737, were included as controls. Interestingly, the reported Bcl-xL picomolar inhibitors, ABT-263 and ABT-737, did not inhibit HBV DNA replication in HepG2.2.15 cells (Fig. 6a, Supplementary Fig. 4a), although both compounds induced significant proteolytic activation of caspase-3, the execution caspase (Fig. 6b,

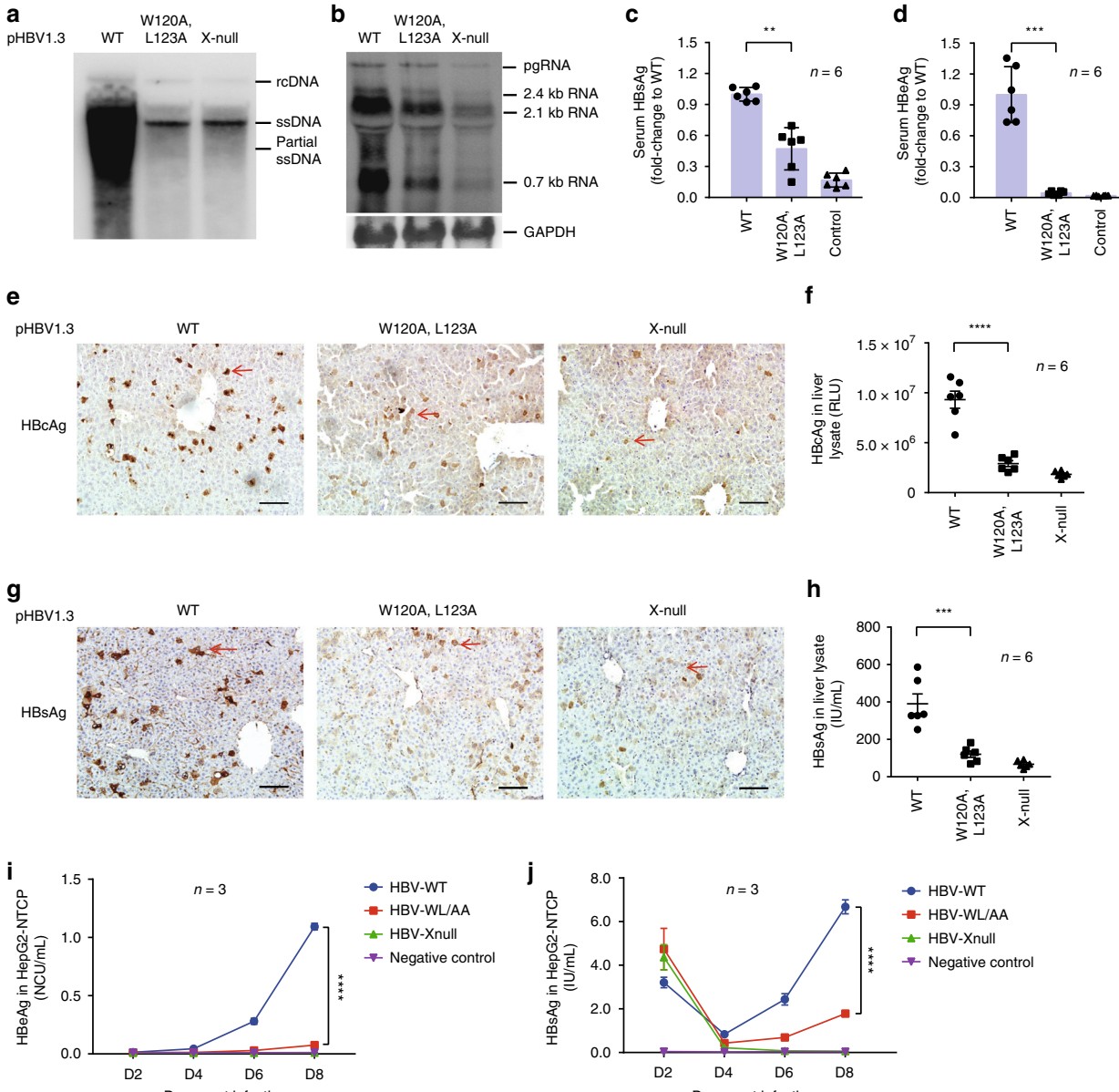

**Fig. 4** HBx Trp120 and Leu123 residues are critical for HBV production in mice and HepG2-NTCP cell. **a**, **b** Southern blot analysis of HBV DNA replication (**a**) and Northern blot analysis of HBV transcription (**b**) in HepG2 cells transfected with pHBV1.3-WT (wild type), pHBV1.3-HBx(W120A/L123A), or pHBV1.3-HBx-null, respectively. **c**–**h** Analysis of HBV viral protein expression in a HBV mouse model. Expression levels of serum HBV S-antigen (HBsAg) (**c**) and HBV e-antigen (HBeAg) (**d**) and expression levels of intrahepatic HBV core antigen (HBcAg) (**f**) and HBsAg (**h**) in liver lysates from mice 3 days post hydrodynamic injection of the pHBV1.3-WT replicon, the pHBV1.3-HBx(WL/AA) replicon, or the pHBV1.3-HBx-null replicon were analyzed by chemiluminescent enzyme immunoassay (CLEIA), respectively. The expression levels of intrahepatic HBcAg (**e**) and HBsAg (**g**) in the liver tissue sections from these mice were also examined by immunostaining. Scale bar is 100 μm. The red arrows indicate the hepatic cells with HBcAg or HBsAg expression. pcDNA3.1-EGFP was co-injected with the HBV replicon as an injection marker to normalize the transfection efficiency ($n = 6$ biologically independent animals). **i** Kinetics of HBeAg and **j** HBsAg levels post HBV-WT (blue line), HBV-WL/AA (red line), and HBV-Xnull (green line) infection in HepG2-NTCP cell line, supernatant of untreated HepG2-NTCP cell was used as a negative control (purple line) ($n = 3$ biologically independent samples). The data represent mean ± SD. Significant differences between groups are indicated on the top. **$P < 0.01$, ***$P < 0.001$, ****$P < 0.0001$; two-tailed unpaired $t$-tests. Source data are provided as a Source Data file

Supplementary Fig. 4b). By contrast, the HBx-aa118–127 peptide, which contains the secondary structure observed in the Bcl-xL/HBx-aa113–135 structure, significantly inhibited HBV replication and HBsAg expression in HepG2.2.15 cells (Fig. 6c, d). On the other hand, HBx-aa113–135, the longer peptide that could rescue defective HBV replication and viral gene expression from the pHBV1.3-Xnull replicon, did not inhibit HBV replication or HBsAg expression in HepG2.2.15 cells (Fig. 6c, d). We also used

HBV-infected HepG2-NTCP cells to evaluate the inhibitory effect of the HBx-aa118–127 peptide on HBV replication, by treating the cells with the peptide at 12.5, 50, and 200 μg/ml concentrations, respectively. Significant, dose–response suppression of the HBeAg expression and HBV DNA levels were observed (Fig. 6e, f), with no obvious cytotoxicity detected (Fig. 6g). Because the HBx-aa118–127 peptide constitutes the short two-helical-turn amphipathic α-helix structure observed in the Bcl-xL/HBx-

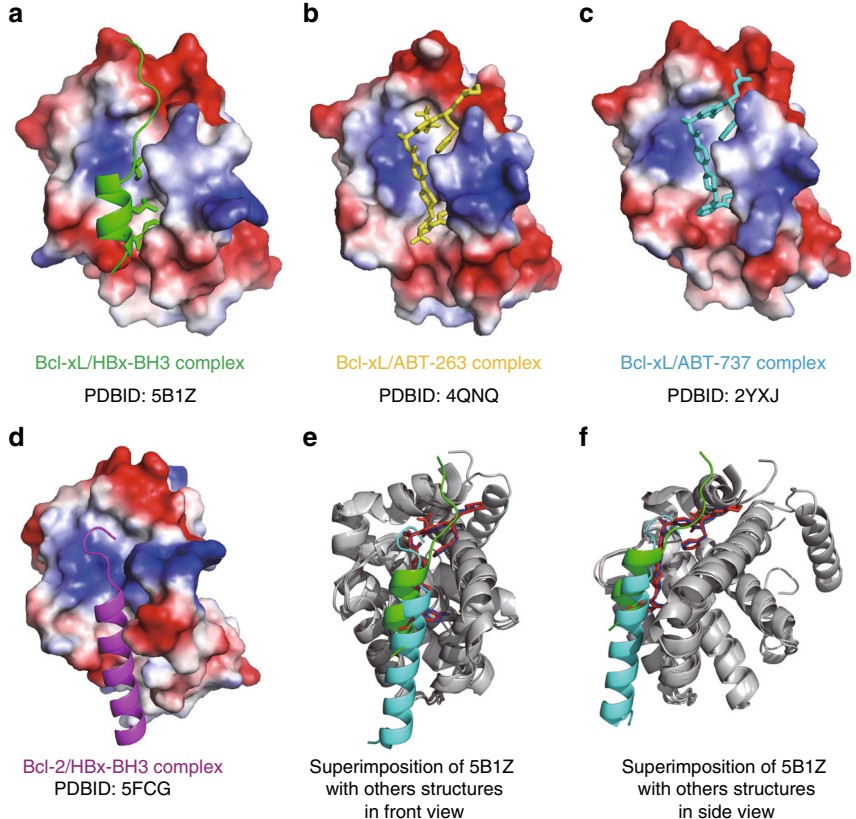

**Fig. 5** Comparison of Bcl-xL/HBx-aa113–135, Bcl-xL/ABT-263, Bcl-xL/ABT-737, and Bcl-2/HBx-aa110–135 structures. **a** Cartoon structure of Bcl-xL in complex with HBx-aa113–135 (PDB ID 5B1Z). **b** Cartoon structure of Bcl-xL in complex with ABT-263 (PDB ID 4QNQ). **c** Cartoon structure of Bcl-2 in complex with ABT-737 (PDB ID 2YXJ[23]). **d** Cartoon structure of Bcl-2 in complex with HBx-aa110–135 (PDB ID 5FCG[8]). The structures of Bcl-xL and Bcl-2 are in surface mode, the HBx-BH3-like motifs are in cartoon mode, and key residues and ABT molecules are indicated in stick mode. Notably, compared with the Bcl-xL/HBx-aa113–135 structure, the HBx-BH3-like helix in the Bcl-2/HBx-aa110–135 structure is shifted ~8 Å towards the N-terminus. Hence, the side chain of the crucial Trp120 residue is pointing to the opposite direction and more than 10 Å away from the conserved hydrophobic pocket formed by the side chains of Phe105, Leu108, Val126, and Phe146 that is critical for binding the HBx-BH3-like motif. **e, f** The structural superimposition (front and side views) of Bcl-xL/HBx-BH3-like complex, Bcl-xL/ABT-263 complex, Bcl-xL/ABT-737 complex, and Bcl-2/HBx-BH3-like complex. Bcl-xL and Bcl-2 are colored in dark gray. The HBx-BH3-like peptide in the Bcl-xL/HBx-BH3-like complex, ABT-263, ABT-737, and the HBx-BH3-like peptide in the Bcl-2/HBx-BH3-like complex are colored in green, red, blue, and cyan, respectively

aa113–135 structure, these results suggest that compared with HBx-aa113–135 the HBx-aa118–127 peptide is sufficient to mediate binding to Bcl-xL, but loses the ability to promote HBV replication, leading to interference with binding of viral HBx to Bcl-xL in HepG2.2.15 cells and inhibition of HBV replication and transcription in HepG2-NTCP cells. Hence, the HBx-aa118–127 peptide could serve as the initial drug lead in peptidomimetics design to develop more potent and specific HBV inhibitors.

Importantly, the Bcl-xL binding pockets for HBx-aa118–127 and ABT-263/ABT-273 are distinct. The Bcl-xL binding surface for HBx-aa118–127 is a small pocket that encapsulates Trp120 and Leu123 of HBx (Fig. 5a), whereas the Bcl-xL binding pocket for ABT-263 or ABT-273 is a shallow groove accommodating the elongated ABT molecules (Fig. 5b, c). Structural superimposition of Bcl-xL complexes with these small molecules shows that the side chain of Trp120 is well separated from the "chloride-benzene" moiety of ABT molecules and the aromatic rings of the Trp120 side chain and the "chloride-benzene" moiety of the ABT molecules are rotated 90° away from each other (Supplementary Fig. 3b and Fig. 6). Hence, the two binding pockets accommodating the side chain of Trp120 in the HBx-BH3-like motif and the ABT molecules are largely separated. To verify this structural observation, we performed isothermal titration calorimetry (ITC) assays to investigate whether binding of Bcl-xL by ABT molecules

interferes with binding of HBx-aa118–127 to Bcl-xL. We measured the binding affinity between HBx-aa118–127 and Bcl-xL or between HBx-aa118–127 and Bcl-xL presaturated with ABT-263 binding. Our data show that the dissociation constants of the HBx-aa118–127/Bcl-xL complex and the complex of HBx-aa118–127 with Bcl-XL presaturated with ABT-263 are 10.03 ± 2.95 and 20.96 ± 4.05 μM, respectively (Supplementary Fig. 5). The mere 2-fold decrease in binding affinity strongly suggests that the critical binding pockets in Bcl-xL for HBx-aa118–127 and ABT molecules are distinct.

In parallel, we investigated whether ABT-263 could interfere with the inhibitory effect of HBx-aa118–127 on HBV replication. HepG2.2.15 cells were treated with HBx-aa118–127 alone, ABT-263 alone, HBx-aa118-127 and ABT-263 together, or ABT-263 followed by HBx-aa118–127. Consistent with our previous observations, HBx-aa118–127 inhibited and ABT-263 did not affect HBV replication (Fig. 7). Interestingly, treatment of HBx-aa118–127 and ABT-263 together or ABT-263 treatment followed by HBx-aa118–127 treatment both resulted in similar levels of inhibition on HBV replication and viral gene expression to those by HBx-aa118–127 alone (Fig. 7). These results provide strong support to our findings that ABT-263 and HBx-aa118–127 bind to two separated pockets in Bcl-xL and could bind to Bcl-xL simultaneously.

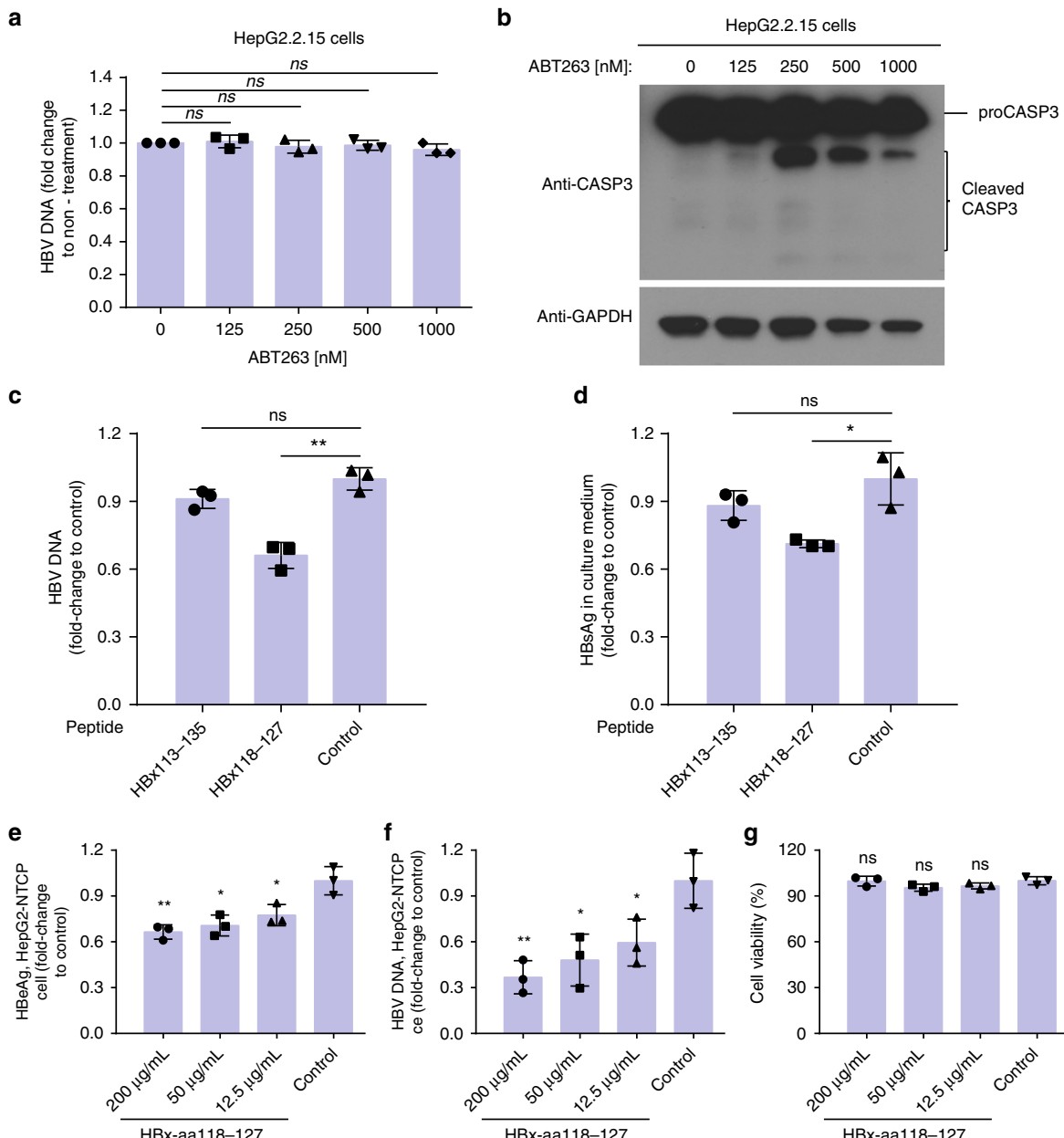

**Fig. 6** The HBx-aa118–127 peptide, but not ABT-263, inhibits HBV replication and transcription in HepG2.2.15 cells. **a** HepG2.2.15 cells were treated with 125, 250, 500, and 1000 nM of ABT-263 for 2 days. Cytoplasmic HBV viral particles were isolated and the viral DNA replication intermediates were quantified (Methods). The results represent the fold change of the replicative intermediates in HepG2.2.15 cells from the indicated treatments, compared with those without treatment. **b** HepG2.2.15 cells were treated with the indicated concentrations of ABT-263 for 2 days. The lysate was analyzed by immunoblotting using antibodies to caspase-3 (CASP3) and GAPDH (glyceraldehyde 3-phosphate dehydrogenase). **c**, **d** The HBx-aa113–135, HBx-aa118–127, or the control peptides were delivered into HepG2.2.15 cells using PEP-1 nanoparticles (Methods). The levels of HBV DNA (**c**) and the levels of HBV S-antigen (HBsAg) expression (**d**) in culture medium collected from HepG2.2.15 cells were measured (Methods). The results are shown as fold change compared to that of the control peptide. **e**–**g** HepG2-NTCP cells were transfected with HBx118–127 peptide with concentration of 12.5, 12.5, and 50 μg/ml or unrelated control peptide during HBV infection (multiplicity of infection (MOI) = 200), respectively. **e** HBV e-antigen (HBeAg), **f** HBV DNA, and **g** cell viability were measured at day 8 after HBV infection. Data shown represent mean ± SD. Significance is indicated on the top. Not significant (ns): *$P$ < 0.05, **$P$ < 0.01; two-tailed unpaired $t$ tests (n = 3 biologically independent samples). Source data are provided as a Source Data file

## Discussion

More than 248 million people worldwide are chronically infected by HBV and are at a high risk of suffering from HBV-induced liver disease and HCC[25,26]. The HBx protein encoded by the HBV viral genome is consistently over-expressed in the livers and tumors of HBV chronic carriers and is involved in the pathogenic development of hepatocytes[27–30]. Studies based on HepaRG cells[31] and human-liver-chimeric mice[32] have indicated that HBx

is required to initiate and maintain HBV replication. The interactions of HBx with multiple host factors, such as DDB1[3,4], SMC5/6[6], Bcl-2, and Bcl-xL[7,8], and their important roles in HBV replication and viral gene expression, have been reported in a number of studies and reviews[5,31,33]. Hence, HBx is an important therapeutic target, in addition to the viral polymerase that has already been targeted for HBV control and treatment[31,34–37]. The discovery that HBx interacts with two host factors[7,12], Bcl-2 and

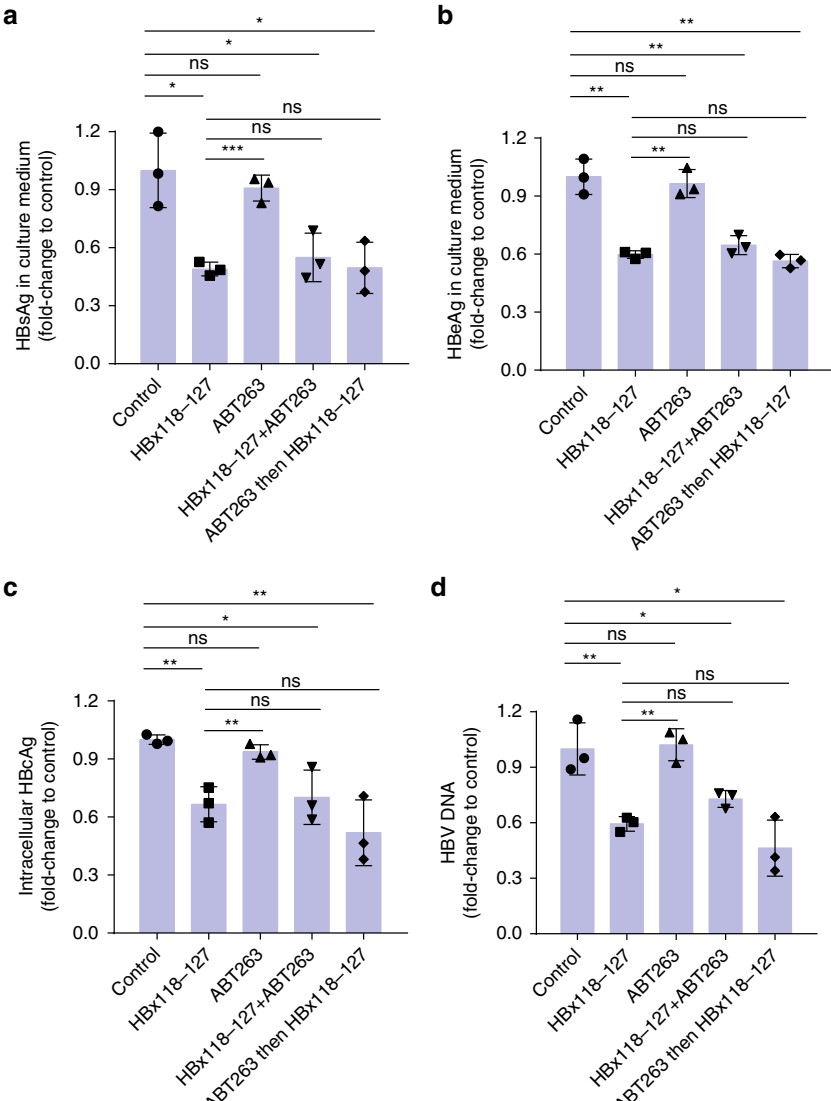

**Fig. 7** ABT-263 does not affect the inhibitory effect of HBx118–127 on HBV replication in HepG2.2.15 cells. **a–d** HepG2.2.15 cells were pre-seeded 12 h before peptide treatment. In all panels, from left to right (bars 1–5), cells were treated with placebo buffer, HBx118–127 only, ABT-263 only, HBx118–127 and ABT-263 together, and ABT-263, followed by HBx118–127, respectively. The cells in bars 1–4 were treated for 48 h, whereas the cells in bar 5 were first treated by ABT-263 for 12 h, followed by treatment with HBx118–127 for another 36 h. The final concentration of ABT-263 was 1 μM in the culture medium, while HBx-aa118–127 was delivered by PEP-1 nanoparticles. After 48-h treatment, the protein expression levels of HBV S-antigen (HBsAg) (**a**) and HBV e-antigen (HBeAg) (**b**) in the culture medium, the intracellular protein expression level of HBV core antigen (HBcAg) (**c**), and the HBV viral DNA tilter (**d**) in the culture medium of HepG2.2.15 cells were measured (Methods). The results are shown as fold changes compared to the values obtained from the cells treated with the placebo buffer. Data shown represent mean ± SD. Significance is indicated on the top. Not significant (ns): *$P < 0.05$, **$P < 0.01$, ***$P < 0.001$; two-tailed unpaired $t$ tests ($n = 3$ biologically independent samples). Source data are provided as a Source Data file

Bcl-xL, through its BH3-like motif to induce increase of inter-cellular calcium required for HBV viral replication suggests that therapeutic intervention of the interactions between HBx and Bcl-2/Bcl-xL proteins could be a new option to treat chronic HBV.

In this study, we report the crystal structure of a HBx-BH3-like peptide in complex with the Bcl-xL protein. Notably, three residues in the HBx-BH3-like motif, Trp120, Leu123, and Ile127, locate on one side of a short α-helix and snuggle into a novel BH3-binding pocket in Bcl-xL through multiple hydrophobic interactions among side chains of both proteins. These hydrophobic interactions are crucial for HBx binding to Bcl-xL and for HBV viral replication and transcription in hepatocytes. Substitutions of two of these residues, Trp120 and Leu123, with alanine disrupt HBx binding to Bcl-xL in vitro and significantly reduce HBV viral replication and transcription in vivo. These

observations raise the possibility that small chemical compounds or peptides can be designed or screened for to reduce or block the interactions between Bcl-xL/Bcl-2 and HBx, leading to inhibition of HBV replication.

In prior studies, two BH3-only mimetics, ABT-263 and ABT-737, were reported as potent inhibitors for Bcl-xL, Bcl-2, and Bcl-w[38,39]. These two small molecules were designed to mimic the action of the BH3-only proapoptotic proteins, binding with high affinity to anti-apoptotic Bcl-2 and Bcl-xL proteins to induce apoptosis of treated cancer cells. As such, they could potentially be used as HBx-BH3-like mimetics to inhibit HBV replication and viral gene expression. However, our results show that ABT-263 and ABT-737 do not inhibit HBV replication in HepG2.2.15 cells (Fig. 6a, Supplementary Fig. 4a), although both could significantly induce proteolytic activation of the key apoptotic

caspase, caspase-3 (Fig. 6b, Supplementary Fig. 4b), indicating that they cannot act as HBx-BH3-like mimetics. By contrast, our HBx-BH3-like mimetic, the short HBx-aa118–127 peptide, significantly inhibits HBV replication and transcription in HepG2.2.15 cells (Fig. 6c, d). These results are consistent with the observations from the superimposition of the Bcl-xL/HBx-BH3-like complex structure with the Bcl-xL/ABT-263 and Bcl-xL/AGBT737 complex structures, which clearly reveal that two different surface pockets in Bcl-xL mediate binding with the HBx-BH3-like motif and with ABT-263 or ABT-737, respectively (Fig. 5a–c and Supplementary Fig. 6). Although a slight overlap between the N-terminal side chains of the bound HBx peptide at residues 120–121 with the "chloride-benzene" moiety of the bound ABT-263 may occur (Supplementary Fig. 6), the Bcl-xL structure appears to be quite flexible, as demonstrated by its numerous slightly different structures in complex with different bound partners[13–15], and could slightly alter its conformation upon binding one BH3 partner to accommodate binding of another different partner. Indeed, ITC assays show that prior binding of ABT-263 to Bcl-xL only marginally affects binding of HBx-aa118–127 to Bcl-xL (Supplementary Fig. 5), supporting the notion that the two binding pockets in Bcl-xL for HBx-aa118–127 and ABT-263 are largely separated. Moreover, prior binding of ABT-263 to Bcl-xL does not affect the ability of HBx-aa118–127 to inhibit HBV viral replication and gene expression (Fig. 7). These results strongly suggest that the two binding pockets in Bcl-xL for HBx-aa118–127 and ABT-263 are separated and could accommodate binding of both molecules with minimal interference.

Recently, a somewhat similar Bcl-2/HBx-BH3-like peptide complex structure was reported[8], in which Bcl-2 adopts a similar structure to that of Bcl-xL in the Bcl-xL/HBx-aa113–135 complex. However, the bound HBx-BH3-like peptides in these two structures, which differ by only three residues, display rather different secondary structures and critical side-chain orientations and interactions[8]. Compared with the Bcl-xL/HBx-aa113–135 complex structure, the HBx-BH3-like helix in the Bcl-2/HBx-aa110–135 structure is shifted ~8 Å towards the N-terminus. Hence, the side chain of the Trp120 residue in the Bcl-2/HBx-aa110–135 structure is more than 10 Å away from the conserved hydrophobic pocket identified in Bcl-xL that accommodates the side chain of the Trp120 residue (Fig. 5a, d). In the Bcl-2/HBx-aa110–135 structure[8], the HBx-BH3-like motif (HBx-aa110–135) binds to a nonconserved shallow groove formed by Gly89/Cys90 and the importance of HBx binding to this nonconserved groove in HBV replication and transcription has not been tested. By contrast, the new hydrophobic pocket in Bcl-xL that we discovered is physiologically important, because substitutions of Trp120 and Leu123, whose side chains interact with this pocket, disrupted the HBx/Bcl-xL interaction (Fig. 2) and HBV viral replication and transcription in HepG2 cells, in HepG2-NTCP cells, and in mouse hepatocytes (Figs. 3 and 4). Moreover, a shorter HBx-aa118–127 peptide designed based on the Bcl-xL/HBx-aa113–135 structure not only significantly inhibited HBV replication and transcription in HepG2.2.15 cells (Fig. 6c, d) but also exhibited similar inhibitory effects in a HBV infection system based on HepG2-NTCP cells (Fig. 6e, f). These results together provide strong supporting evidence that the molecular interactions observed in the Bcl-xL/HBx-aa113–135 structure are physiologically significant.

As a multi-functional viral protein, HBx has been shown to interact with a number of host factors[3–8], including DDB1, Bcl-2 and Bcl-xL. The W120A/L123A mutations in the HBx-BH3-like motif could potentially affect HBx interactions with other host factors. One with particular interest is the DDB1 protein, a component of the DDB1-CUL4-ROC1 E3 ligase complex, which

interacts with HBx to target Smc5/6 for ubiquitylation and degradation[6,9], which relieves inhibition of HBV virus gene expression by Smc5/6. Further studies are needed to assess the impact of the HBx-BH3-like peptide and relevant mutations on DDB1-mediated Smc5/6 degradation. All the data presented in this study strongly support the model that HBx directly targets Bcl-xL to promote HBV replication and virus gene expression in cultured HepG2 cells, in a simplified HBV cell infection system (HepG2-NTCP cells), and in a mouse HBV model. It remains to be verified if this HBx/Bcl-xL interaction is crucial for HBV infection, replication and gene expression in human livers. We also could not rule out the possibility that additional host targets might interact with this BH3-like domain to impact HBV life cycle.

Structural comparison of Bcl-xL in complex with other proteins containing the BH3 domain, such as Beclin-1, Bad, Bim and Bak, shows that the structure of Bcl-xL could be rather flexible to accommodate binding to various proteins with BH3 motifs that have slightly different compositions, lengths, and side-chain properties[14,40,41]. Notably, the HBx-BH3-like motif is a noncanonical BH3 motif, in which the conserved LXXXGD motif seen in canonical BH3 domains is replaced by the WXXXGE motif. Moreover, the HBx-BH3-like motif adopts a short, two-helical-turn α-helix conformation instead of a canonical, longer three-helical-turn α-helix seen in typical BH3 domains. Compared with other BH3 domains in complex with Bcl-xL[9,10,31,32], the HBx-BH3-like motif slides ~2 Å off the canonical BH3-binding pocket of Bcl-xL, which allows the bulky Trp120 side chain to snuggle into a different pocket formed by the side chains of Phe105, Leu108, Val126, and Phe146 in Bcl-xL (Fig. 1c). Hence, ABT-263 or ABT-737 derivatives containing an extended arm mimicking the side chain of Trp120 could be designed to occupy the Trp120-binding pocket in Bcl-xL and to enable the ABT derivatives to block HBx binding to Bcl-xL, leading to better suppression of HBV DNA replication and viral protein production. This is significant because the disease progression in chronic hepatitis B patients correlates with the level of HBV production[42–44].

Importantly, the HBx-BH3-like motif not only is necessary for HBV viral replication and transcription but also is sufficient to promote HBV replication and transcription in hepatic cells, when supplied in trans to a replication-defective, HBx-null HBV replicon. The HBx-aa113–135 peptide, delivered into the HepG2 cells through PEP-1 nanoparticles, is capable of rescuing the viral replication and transcription defects of the HBx-null HBV replicon (Fig. 3). To our knowledge, this is the first demonstration that a short peptide is capable of promoting HBV replication and transcription in hepatic cells. These findings not only underscore the importance of the HBx-BH3-like motif to the HBV life cycle but also promise potential interference of HBV replication and transcription through BH3-like peptidomimetics.

In summary, our structural and functional analyses of the HBx-BH3-like domain in complex with Bcl-xL reveal the molecular principles of HBx-mediated HBV replication and transcription and provide much-needed structural insights to guide development of small chemical compounds and peptidomimetics that can target the Bcl-xL/HBx interaction to inhibit HBV replication and transcription and to treat HBV-induced liver diseases.

## Methods

**Expression and purification of Bcl-xL.** The codon-optimized DNAs encoding Bcl-xLΔTM (the flexible loop, residues 27–82, and the transmembrane domain, residues, 197–233, are deleted, total 153 residues, named as Bcl-xL thereafter) was ordered from Genewiz (www.genewiz.com). The recombinant Bcl-xL with N-terminal SUMO-fused 6× His tag was generated by the insertion of the PCR

products into pETSUMO vector (Invitrogen; www.thermofisher.com) and verified by sequencing. Recombinant proteins were expressed in *Escherichia coli* (BL21/DE3) strain overnight at 20 °C and proteins were induced by 0.4 mM isopropyl β-D-thiogalactoside. Cells were harvested by centrifugation and the pellets were re-suspended and passed through a cell disruptor (www.avestin.com) three times. After ultracentrifugation at 40,000 rpm for 1 h, the supernatant for His-tagged recombinant proteins were purified through Ni$^{2+}$ affinity column, cleaved by Ulp-1, followed by HiLoad Superdex S-75 26/60 column (GE Healthcare). The purified proteins were dialyzed against stabilization buffer containing 20 mM Tris-HCl (pH 7.0) and 100 mM NaCl and concentrated to 10–15 mg/ml in a Centriprep-30 (Amicon) for subsequent crystallization and biochemical analysis.

**Peptide synthesis**. The HBx peptides were ordered from Invitrogen, Shanghai (China) by standard solid-phase synthesis method. A total of five peptides were synthesized, including HBx-aa113–135 (KDCVFKDWEELGEEIRLKVFVLG), C-terminal biotinylated HBx-aa113–135-biotin ((KDCVFKDWEELGEEIRLK VFVLG-K-biotin), C-terminal biotinylated HBx-aa113–135-(W120A/L123A double mutation) (KDCVFKDAEEAGEEIRLKVFVLG-K-biotin), HBx-aa118–127 (KDWEELGEEI), pep-biotin (SSTTSTGPCKTCTTPAQGTSMFP-K-biotin) (equal-length control biotin-labeled peptide), and PEP-1 (KETWWETWWTEW SQPKKKRKV) (transfection accessory peptide).

**Crystallization and data collection**. Peptide HBx-aa113–135 was dissolved in a buffer containing 10 mM carbonic acid (pH 9.5) and 10 mM NaCl at a concentration of 10 mg/ml, and then mixed with Bcl-xL at a final molar ratio of 2:1 (peptide:protein). Crystals of Bcl-xL in complex with HBx-aa113–135 were grown at 20 °C by a mixture of 1.0 μl of protein with 1.0 μl of reservoir containing 1.05 M sodium citrate and 0.1 M CHES (2-(*N*-cyclohexylamino)ethane- sulfonic acid) (pH 9.5). These crystals grew to a maximum size of 0.3 mm × 0.3 mm × 0.1 mm over the course of 5 days. These crystals belong to the space group $C222_1$ and contain two Bcl-xL molecules and two HBx-aa113–135 molecules per asymmetric unit. The unit cell dimensions are $a = 50.71$ Å, $b = 135.32$ Å, and $c = 97.59$ Å.

These crystals were flash frozen (100 K) in the reservoir solution supplemented with 30% (v/v) glycerol. One-wavelength data set (total 180° with 1° oscillation) were collected at home X-ray source at wavelength 1.5418 Å and processed by HKL2000 (www.hkl-xray.com).

**Structural determination**. The structure of Bcl-xL in complex with HBx-aa113–135 was determined by molecular replacement using the program MOL-REP/CCP4 (www.ccp4.ac.uk) and coordinate from human Bcl-xL crystal structure (accession code 1R2D). The model was built by using the program O (http://xray.bmc.uu.se/alwyn) and refined using REFMAC/CCP4 (www.ccp4.ac.uk) to 2.15 Å. The final refined model of Bcl-xL/HBx-BH3 comprises residues 1–26 and 83–200 of Bcl-xL and residues 118–135 of HBx-aa113–135. The residues aa113–117 of HBx-aa113–135 and the C-terminal nine residues of Bcl-xL are disordered and omitted from the model. The crystallographic statistic details of these structures are listed in Table 1.

**HBx plasmids construction**. The DNA fragment encoding HBx was amplified from HBV genome by PCR and subcloned into a modified pTT22 backbone (pTT22m) via its *Bam*HI and *Hin*d III sites, in which the fragment of puromycin gene in pTT22 vector was replaced by mCherry. The HBx mutant constructs containing W120A and L123A mutations were generated using a QuickChange Site-Directed Mutagenesis Kit (Stratagene) and verified by DNA sequencing. pHBV1.3 replicon containing 1.3-fold of the whole HBV genome was constructed in pGEM-4Z vector, pHBV1.3 containing HBx (W120A, L123A) and pHBV1.3-HBx-null without HBx expression were made by QuickChange Site-Directed Mutagenesis Kit (Stratagene), and verified by DNA sequencing.

**Cell lines**. HepG2 cells were purchased from the American Type Culture Collection (#HB-8065). HepG2.2.15 cells were the kind gift of Prof. Xu Lin (Fujian Medical University, China). HepG2-NTCP cell line was generated by a sleeping-beauty transposon-based system that co-expresses fluorescent protein mCherry and puromycin through all-in-one plasmid transfection[45].

**Antibodies**. The following antibodies were used in this study: anti-HBx antibody 16F9 (1:1000; developed and validated by Prof. Xia's lab). Anti-Bcl-xL antibody (1:2000; Proteintech, #a10783-1-AP). Anti-β-tublin antibody (1:2000; Abcam, #ab179513). Anti-caspase-3 antibody (1:1000; Proteintech, #19677-1-AP). Anti-GAPDH (glyceraldehyde 3-phosphate dehydrogenase) antibody (1:5000; Proteintech, #60004-1-Ig). Horse radish peroxidase (HRP)-conjugated goat anti-mouse IgG (1:5000; Proteintech, #SA00001-1). HRP-conjugated goat-anti-rabbit IgG (1:5000; Thermo Fisher Scientific, #65-6120).

**Immunoprecipitation assay**. HepG2 cells were lysed in lysis buffer [100 mM NaCl, 0.5 mM MgCl$_2$, 0.15 mM CaCl$_2$, 1% (v/v) Nonidet P-40, 10 mM Tris-HCl, pH 8.0] containing protease inhibitor mixture tablets (Roche). Cell lysate was centrifuged at 12,000 × *g* for 10 min at 4 °C to removed debris and precleared with

**Table 1 Data collection and refinement statistics**

| Data collection | |
|---|---|
| Protein target | Bcl-xL in complex with HBx-BH3 |
| Space group | $C222_1$ |
| PDB ID | 5B1Z |
| Cell dimensions *a*, *b*, *c* (Å) | 50.71, 135.32, 97.59 |
| Wavelength (Å) | 1.5418 |
| Resolution (Å)[a] | 50–2.15 (2.19–2.15) |
| $R_{sym}$ (%) | 8.9 (41.9) |
| $I/\sigma(I)$ | 22.4 (4.4) |
| Completeness (%)[a] | 99.9 (99.9) |
| Redundancy | 6.9 (6.5) |
| Search model | 1R2D |
| **Refinement** | |
| Resolution range (Å) | 50–2.15 (2.21–2.15) |
| No. of reflections | 17,371 |
| $R_{work}$ ($R_{free}$) (%) | 20.1/27.7 (23.3/36.4) |
| No. of atoms | |
| Protein | 2614 |
| Water | 151 |
| *B*-factors (Å$^2$) | |
| Protein | 31.8 |
| Water | 28.2 |
| R.m.s. deviations | |
| Bond lengths (Å) | 0.015 |
| Bond angles (°) | 1.654 |
| % favored (disallowed) in Ramachandran plot | 94.4 (0) |

[a]Values for the highest-resolution shell are within parentheses

magnetic beads conjugated with streptavidin (Dynabeads® M-280 Streptavidin, Invitrogen). Subsequently, the precleared supernatant was incubated with C-terminal biotinylated peptides for 1 h with gentle shaking at 4 °C. A new batch of streptavidin beads were added and incubated for another 2 h. The beads were washed five times with the lysis buffer and the bound proteins were resolved on a 13.5% sodium dodecyl sulfate-polyacrylamide gel and detected by anti-Bcl-xL antibody (Cell Signaling Technology).

Similarly, HepG2 cells transfected with pTT22m-HBx-WT or pTT22m-HBx-(W120A/L123A) plasmid were lysed. Co-IP experiments were performed using an anti-HBx antibody (16F9) and Dynabeads Protein G for Immunoprecipitation (Thermo Fisher Scientific, catalog no. 10004D).

**Peptide transfection assay**. HepG2 cells were grown in Dulbecco's modified Eagle's medium (DMEM) with 10% Gibco fetal bovine serum. C-terminal biotinylated-labeled HBx peptides were transfected into HepG2 cells by pep-1[16], a peptide carrier for the delivery of biologically active proteins into mammalian cells. HBx peptides and pep-1 were firstly dissolved in DMEM to 1 mg/ml, then adjusted to 500 μl DMEM to a final concentration of 100 μg/ml, and incubated at 37 °C for 30 min in the tube. The incubated peptide mixture was added into pre-seeded HepG2 cells in a 24-well plate and incubated for another 30 min, and then washed five times by phosphate-buffered saline (PBS) to remove the free peptides. The transfected C-terminal biotinylated HBx peptides were detected by using streptavidin-FITC (fluorescein isothiocyanate) equipment (Thermo Fisher Scientific) following the standard immunostaining methods.

**HBV production rescue assay**. Transfection of plasmids into HepG2 cells was carried out using X-tremeGENE HP DNA Transfection Reagent (Roche), following the manufacturer's protocol. A transfection efficiency of 20–30% could be achieved. Cells were pre-seeded 24 h before transfections. pHBV1.3-HBx-null plasmid was transfected into HepG2 cells for 6 h, then HBx-aa113–135, HBx-aa113–135-(W120A, L123A), or control peptide was transfected by pep-1 carrier and incubated for additional 48 h without the removal of peptides. The culture media of the HepG2 cells were harvest for analysis.

Similarly, the culture media of the HepG2 cells co-transfected by pHBV1.3-HBx-null and pTT22m-HBx-WT, pHBV1.3-HBx-null and pTT22m-HBx-(W120A, L123A), or pHBV1.3-HBx-null and pTT22m control were harvested for analysis.

To evaluate the potential inhibitory effects of peptides on HBV replication and transcription in HepG2.2.15 cells, HepG2.2.15 cells were pre-seeded 24 h before peptides treatment. HBx-aa113–135, HBx-aa118–127, or control peptide was transfected through pep-1 carriers and incubated for additional 48 h without the removal of peptides. The culture media of HepG2.2.15 cells were then harvested for analysis.

**HBV infection in HepG2-NTCP cell line**. To obtain the recombinant HBV with or without HBx-WL/AA double mutation, Huh7 cells was transfected with pHBV1.3-WT, pHBV1.3-HBx-(W120A/L123A), and pHBV1.3-HBx-null, respectively. The culture medium was harvested at 3, 5, 7, and 9 days post transfection and concentrated by 25% sucrose ultracentrifugation. The sediment was re-suspended with serum-free DMEM. Virus were normalized by HBV DNA quantification. HepG2-NTCP cell was seeded in 48-well plate, and then infected with HBV-WT, HBV-WL/AA, and HBV-Xnull virus, three repeated wells for each virus, respectively. Multiplicity of infection was equal to 200; the culture medium was harvested and replaced every 2 days for detection.

**ABT-263 and ABT-737 treatment**. ABT-263 and ABT-737, the picomolar BH3 mimetics, were purchased from Apexbt Technology LLC (Houston, TX, USA). HepG2.2.15 cells were treated with 125, 250, 500, and 1000 nM of ABT-263 or ABT-737 for 48 h. Cytoplasmic HBV viral particles and HBV DNA were then isolated for quantification.

**Isothermal titration calorimetry**. Microcalorimetry was used for all measurements. Bcl-xL protein was prepared at a final concentration of 10 μM in a buffer containing 25 mM Tris-HCl (pH 8.0). The ABT-263 was dissolved in the same buffer to a final concentration of 500 μM, while the peptides were dissolved to a final concentration of 800 μM. All experiments were carried out at 25 °C. Data were analyzed using the software Origin (OriginLab).

**Animal studies**. All mice were maintained under specific pathogen-free conditions in the Laboratory Animal Center of Xiamen University. The experiments were conducted under approval of the Institutional Animal Care and Use Committee at Xiamen University (XMULAC20150016) and were in accordance with the Guide for the Care and Use of Laboratory Animals.

C57BL/6 mice (male, 6–7 weeks old) were divided into three groups (six mice for each injection group). Twenty micrograms of the pHBV replicon (pHBV1.3-WT, pHBV1.3-HBx-(W120A/L123A), and pHBV1.3-HBx-null) plus 2 μg of pcDNA3.1-SEAP were injected into the tail veins of mice within 4 s in a volume of PBS equivalent to 10% of the mouse body weight[20]. Livers of these mice were collected and assayed for the levels of HBcAg and HBsAg by immunohistochemical staining at 2 days after injection.

**HBV protein, RNA and DNA analysis**. The levels of HBsAg, HBeAg, and HBcAg expression were measured by chemiluminescence, using Commercial Assay Kit (Wantai, Beijing, China). Northern blot analysis of HBV RNA, Southern blot analysis of HBV DNA, and real-time PCR quantification of HBV DNA are described in the Supplementary Methods.

**Reporting summary**. Further information on research design is available in the Nature Research Reporting Summary linked to this article.

## Data availability

All data in this study are available from the corresponding author upon reasonable request. The coordinate has been deposited in the Protein Data Bank, www.pdb.org, with accession number 5B1Z (Bcl-xL in complex with HBx-BH3 motif). A reporting summary for this article is available as a Supplementary Information file. The source data for Figs. 2a, b, 3b–i, 4a–d, f, h–j, 6a–g, and 7a–d and Supplementary Fig. 4a, b are provided as a Source Data file.

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

## Acknowledgements

This work was supported by China National Scientific and Technological Major Project (2017ZX10202203-001, 2018YFA0102900), Singapore National Science Foundation (NRF2016NRF-NSFC002-019), the National Natural Science Foundation of China (81571990, 81702006, 81672023), the China Postdoctoral Science Foundation (2017M610398), and the Excellent Youth Foundation of Fujian Scientific Committee (2015J06018). We would like to dedicate this paper to the memory of Prof. Y. Adam Yuan, our beloved colleague, who did not see its completion.

## Author contributions

T.-Y.Z., Y.A.Y., D.X., and N.-S.X. designed the experiments and wrote the article. T.-Y.Z., H.-Y.C., J.-L.C., H.-L.X., X.-B.M., T.-L.L., X.-Z.K., J.-H.Z. and X.-Z. performed the experiments and collected the data. B.Y., C.-H.H., and Q.Y. contributed to data analyses and material support. Study supervision: Y.A.Y., Q.Y., D.X., and N.-S.X.

## Additional information

**Competing interests:** The authors declare no competing interests.

