## [Peer Review File · Nature Communications]

Reviewers' Comments:

Reviewer #1:

Remarks to the Author:

This is an interesting and well designed study that aims to understand the structural and functional relevance of Hbx interaction with the pro-survival Bcl-2 protein Bcl-xL. The manuscript is logically structured, and features a series of compelling experiments that in their totality demonstrate that HBx binding to Bcl-xL is physiologically significant for HBV replication and that this interaction is a suitable drug target due to the observation of a novel interaction site that is unique to the HBx : Bcl-xL interface.

Introduction:

"However, the structural basis of the interaction between the HBx BH3-like motif and the Bcl-2 or Bcl-xL protein and the impact of this interaction to the life cycle of HBV are poorly understood." This is not correct – the crystal structure of HBx and Bcl-xL has been determined (by Jiang et al PNAS, and is discussed in the manuscript at a later point), and should be referenced here in an amended statement that reflects the publication of that paper.

Description of binding interface:

"Moreover, a hydrogen bond is formed between the side chains of Leu194 in Bcl-xL and Asp130 in the HBx BH3-like motif..." – that cant be right, the authors should re-examine this section of the interface and accurately describe any interactions involving Asp130 of HBx as well as Leu 194 of Bcl-xL.

Considering the quite different mode of binding of HBx to Bcl-2 and Bcl-xL, the authors must provide a simulated anneal omit map of the HBx density so the reader can judge the quality of the unbiased electron density for the peptide.

The authors should measure the affinity of HBx 118-127, 113-135 and the 113-135 double mutant peptides – these measurements will provide important quantitative information supporting the viral infection assays.

The authors also speculate that "Bcl-xL could accommodate both HBx-aa118-127 and ABT263 at two different binding pockets with minimal interference." This should be tested experimentally, and represents an important experiment to test the proposed mechanism of action. HBx-aa118-127 should display either none or only minimally reduced affinity for Bcl-xL if Bcl-xL is preincubated with ABT-263.

Figure 1 a and b would benefit from the use of trace rather than full ribbon (in fact the figures in the supplemental data are much clearer) – it is difficult to see the individual amino acids, which is further compounded by the unfortunate placement of the residue labels. Helices should be labelled. Panel c – what is the sigma level of the electron density map? What type of map is it? Panels d,e – the red colour for the labels is difficult to see.

Minor point

Bcl-xL is usually denoted with the capital L as subscript, this should be amended throughout the

manuscript.

Figure 6 and 7 – what is meant by the ns and ** in the graphs? If this is related to the statistical analysis this should be clearly stated in the legend.

Figure S3 – what is meant by “ABT263 tips binding pocket”?

Reviewer #2:

Remarks to the Author:

Review

The authors show the structure of the HBx protein BH3 domain bound to a Bcl domain that is substantially different from the structure determined by Jiang et al (PNAS (2016 113, 2074)). They find the peptide site is shifted from the typical binding site and that the peptide they use does not form the typical all helical structure of other BH3 domains. They do not discuss if a longer peptide might give a different structure. They do observe that their peptide precipitates HBx and that a mutant peptide that specifically removes residues critical to the unusual binding site does not. Surprisingly they find their peptide, which can interact with Bcl but presumably not the SMC5/6 complex or the HBV genome, can activate HBV expression – if reproducible, this is a stunning observation. At a later point they show a truncated peptide inhibits HBV replication (I question whether this is more like and HBx- phenotype). They also test the importance of the unique BH3-Bcl interface they constructed a mutant HBx where two residues W120 and L123 are mutated to A. The mutant did not work as well in cells or in mice, but there are few controls to test the basis of this attenuation, so the result is correlative not mechanistic. In a nice bit of chemical biology the authors show that their peptide and two ABT small molecules do not compete for the same Bcl site.

In summary, the authors show a HBx Bcl interaction that is different from the previous structure but do not consider that the rest of the HBx protein may also affect this structure.

They show how a long version of their peptide is HBV stimulatory while a short one appears to be inhibitory. They need to test if the short peptide can also precipitate HBx.

They show that two Bcl-specific small molecules bind via a non-overlapping site and do not interfere with the activity of HBV suppressive 118-127 peptide.

Major points

The description of Bcl-xL is inadequate (p4 and p17). In the expressed protein, is residue 26 connected to 83? Where does residue 153 figure in (p17)? A protein composed of residues 1-26 and 83-197 is 141 residues long. Is the N-terminal Sumo-6His tag removed before crystallization? If so how? Was it visible in the crystal structure?

The description of the BH3-domain binding pocket is inadequate. Top of page 5, the text refers to the movement of $\alpha 3$ relative to $\alpha 4$, but I have to go to supplementary info to identify the helices. Insert a new panel into figure 1. Is the pocket for the HBx peptide smaller? By what measure? Considering the extended C-terminus, is the buried surface area that different? The data in figure 1f do not show the differences in the residues bound. The lower panels of figure 1 do not appear to be in the same orientation. The electrostatic surface appears to be the standard PyMol vacuum surface, but is not identified as such. If the Bim-peptide structure has a different version of Bcl-xL, especially with regard to differences in the beginning and end points of visible peptide, it will change the surface shading, but the change is not meaningful. Clarify the sequence overlap, trim the models the same, and give them the same orientation. How can there be a hydrogen bond from the side chain of Leu194 to Asp130 in

HBx – the Leu side chain is aliphatic.

For figure 2, is it known that the mutant Bcl-xL is properly folded?

Figure 3. Why does delivery of a BH3-like peptide enhance production of virus markers to 2x background? Does this indicate that HBx interaction with Bcl is virus inhibitory? This possibility needs to be discussed.

Literature is not well cited. The references and PDB accession numbers for all other structures used should be in the text and in the figure legends.

I would like to see electron density for the whole BH3 peptide. The use of electrostatic surfaces without adequate labeling and with different protein constructs is confusing. The backbone trace in Fig S1 is cleaner.

Minor points

Pymol uses a non-standard designation of cartoon for ribbon diagrams. It is better to describe ribbon diagrams as such.

Figure 1. panels c, d, e, and f are incorrectly identified

Figure 4. two panels are labeled 'e'. figure organization is difficult

Figure 5. The panels are too small and should be made larger. The panels show ribbon diagram of peptide BH3-like domains or stick figure diagrams of drugs bound to electrostatic surfaces of the Bcl2 domain. The authors have a more confusing description of the figures. In panel a, the Bcl domain is in a different orientation than in the other panels, making comparison difficult.

Figure 6. Labels a and b are not present

Reviewer #3:

Remarks to the Author:

In this manuscript, Zhang et al described structural and functional analyses of binding between HBV X protein and Bcl family of proteins and defined the interacting domains and residues for the two proteins. This interaction was uncovered a few years ago based on the presence of a putative BH3-like domain on HBx and later supported by some functional studies. The structural biology analysis is done well and rigorously, and appears to be convincing in this manuscript. This reviewer, however, questions the premise that HBx indeed possesses a true BH3-like domain. The original homology alignment (Geng, et al, PNAS 2012; Jiang et al, PNAS 2016) between HBx and authentic BH3-domain-containing proteins, like BAX, BAD, etc was at best marginal. The current structural analyses actually indicated that the HBX BH3-like domain binds to a different region from that of the well-known BH3 binding region on Bcl family of proteins. Furthermore, the HBx BH3-like domain does not exert a similar functional effect as that of the known BH3-mimetics, like ABT-263 and ABT-737. Thus, based on structural and functional criteria, this reviewer is not convinced that that HBx indeed contains a BH3-like domain.

On the other hand, this region of HBx is obviously important to the function of HBx by mutagenesis analysis, which had been shown previously by many other reports on the functions of HBx. While a HBx peptide containing this region can bind to Bcl-xL physically in a highly artificial condition, as shown in this paper, it does not mean that such an interaction indeed occurs in a cellular environment. The fact that another peptide with 3 additional amino acids binds to a different region on Bcl-2, raises

concern regarding the physiological nature of this putative binding domain. The key question of what is the true biological target of this region of HBx remains unresolved.

In this and previous studies, the authors have performed some functional studies by using HBV replication system to support the interaction between HBx and Bcl-2 family of proteins. The studies were performed in transfection system, which is prone to artefacts. HBV infectious system, such as HepG2-NTCP, is readily available now and should be used to validate the functional relevance of this interaction. It is well known that knocking down Bcl family of proteins can induce apoptosis and other untoward effects, and possibly causes nonspecific and indirect perturbation of HBV replication. Thus, findings based on knocking down Bcl-2 need to be interpreted with caution.

It is also puzzling that a HBx mimetic on one hand, can restore the replication defect of a HBV mutant without HBx, but on the other hand, inhibits WT HBV replication.

Minor comments:

1. The legend in Fig. 1 was mislabeled with regard to the panels.
2. The discussion should also provide a more balanced view of the field with respect to other important recent findings regarding the functions of Hbx in HBV replication.
3. The concept of using a HBx mimetic to block HBV replication, as shown in Fig. 7 is not convincing. The inhibitory effects on various HBV replication markers, are not very impressive. Should control for viability and effect on unrelated expression constructs.

Reviewers' comments:

Reviewer #1 (Remarks to the Author):

This is an interesting and well-designed study that aims to understand the structural and functional relevance of Hbx interaction with the pro-survival Bcl-2 protein Bcl-xL. The manuscript is logically structured, and features a series of compelling experiments that in their totality demonstrate that HBx binding to Bcl-xL is physiologically significant for HBV replication and that this interaction is a suitable drug target due to the observation of a novel interaction site that is unique to the HBx : Bcl-xL interface.

Response: We thank Reviewer #1 for the overall positive summary and comments of our study and findings.

1. Introduction:

“However, the structural basis of the interaction between the HBx BH3-like motif and the Bcl-2 or Bcl-xL protein and the impact of this interaction to the life cycle of HBV are poorly understood.” This is not correct – the crystal structure of BHx and Bcl-xL has been determined (by Jiang et al PNAS, and is discussed in the manuscript at a later point), and should be referenced here in an amended statement that reflects the publication of that paper.

Response:

We thank the reviewer for the suggestion. In the revised manuscript, we cited the Jiang et al PNAS study in the introduction in page 4: “Although Bcl-2/HBx-BH3-like complex structure was reported recently⁹, the structural basis of the interaction between the HBx BH3-like motif and the Bcl-xL protein and the impact of this interaction to the life cycle of HBV are poorly understood.”

We also want to point out that Jiang et al PNAS paper mainly described the structure of Bcl-2/HBx-BH3-like complex and did not test the impact of this protein interaction, Bcl-2/HBx-BH3-like, in HBV replication and life cycle. In our study, we not only determined

the structure of the Bcl-xL/HBx-BH3-like peptide complex, but also demonstrated the significance of this Bcl-xL/HBx-BH3-like protein interaction in HBV replication and infection.

2. Description of binding interface:

“Moreover, a hydrogen bond is formed between the side chains of Leu194 in Bcl-xL and Asp130 in the HBx BH3-like motif...” – that can’t be right, the authors should re-examine this section of the interface and accurately describe any interactions involving Asp130 of HBx as well as Leu 194 of Bcl-xL.

Response: We thank the reviewer for pointing out this mistake and amended the description to “Moreover, a hydrogen bond is formed between the main chain of Leu194 in Bcl-xL and the side chain of Asp130 in the HBx BH3-like motif...”

Considering the quite different mode of binding of HBx to Bcl-2 and Bcl-xL, the authors must provide a simulated anneal omit map of the HBx density so the reader can judge the quality of the unbiased electron density for the peptide.

Response: Many thanks for the insightful comments. In the revised manuscript, a new supplementary figure (Fig. S1) was included to show the simulated anneal omit map (Fo-Fc map) covering the whole BH3 peptide. The map is displayed with the sigma value contoured at 2.0. We also stated this in the corresponding figure legend.

The authors should measure the affinity of HBx 118-127, 113-135 and the 113-135 double mutant peptides – these measurements will provide important quantitative information supporting the viral infection assays.

Response: We performed additional binding assays to measure the binding affinities of HBx118-127, 113-135 and 113-135-W120A/L123A double mutant peptides with Bcl-xL, our data showed that the dissociation constants between HBx118-127 and Bcl-xL and

between HBx113-135 and Bcl-xL are $10.03 \pm 2.95 \mu\text{M}$ (Fig. S5a) and $26.5 \pm 13.7 \mu\text{M}$ (Fig. 2c), respectively. On the other hand, the interaction between 113-135-W120A/L123A double mutant peptide with Bcl-xL was non-detectable (Fig. 2d). These new data were added to the revised manuscript.

The authors also speculate that “Bcl-xL could accommodate both HBx-aa118-127 and ABT263 at two different binding pockets with minimal interference.” This should be tested experimentally, and represents an important experiment to test the proposed mechanism of action. HBx-aa118-127 should display either none or only minimally reduced affinity for Bcl-xL if Bcl-xL is preincubated with ABT-263.

Response: Thanks for suggesting this good experiment. In the revised manuscript, we performed ITC experiments to compare the binding affinities between HBx118-127 and Bcl-xL and between HBx118-127 and Bcl-xL preincubated with ABT-263 as suggested. Our data clearly demonstrated that the prior binding of ABT263 only marginally affects the binding of HBx-aa118-127 to Bcl-xL (Fig. S5). These results are consistent with the results from the HBV replication inhibitory experiments showing that the prior incubation of ABT263 with HepG2.2.15 cells did not interfere with the inhibitory activity of HBx 118-127 on HBV replication (Fig. 7).

These data, together with the structural comparisons of three different Bcl-xL complexes with HBx-aa118-127, ABT263 and ABT737 (Fig. 5 and Fig. S6), clearly demonstrate that ABT molecules and HBx-BH3-like peptides bind to two separated binding pockets on Bcl-xL. These two binding pockets are not mutually exclusive and both ABT molecules and HBx-like peptides could bind to these two pockets at the same time.

Figure 1 a and b would benefit from the use of trace rather than full ribbon (in fact the figures in the supplemental data are much clearer) – it is difficult to see the individual amino acids, which is further compounded by the unfortunate placement of the residue labels. Helices should be labelled. Panel c – what is the sigma level of the electron density

map? What type of map is it? Panels d,e – the red colour for the labels is difficult to see.

Response: Fig. 1a and 1b show the overall picture of the Bcl-xL/HBx-BH3-like complex compared with the published structure of the Bcl-xL/Bim complex. We would like to highlight that the overall structures of Bcl-xL in these complexes have marginal changes, whereas the structures of the bound peptides have significant differences in their secondary structures. Hence the ribbon presentation could be a good choice. Moreover, the corresponding Fig. 1d and 1e further highlight the structural differences of the bound peptides. The supplementary figure (Fig. S2) provides more details of the key residues along the binding interface. Furthermore, the detailed interface between the HBx peptide and Bcl-xL has been shown in Fig. 1c (simulated anneal omit map with the sigma value contoured at 1.0). We also changed the color of labels of panel d & e to make it clearer in the revised manuscript.

Minor point

Bcl-xL is usually denoted with the capital L as subscript, this should be amended throughout the manuscript.

Response: Many thanks! We have checked throughout the manuscript carefully and made the corrections.

Figure 6 and 7 – what is meant by the ns and ** in the graphs? If this is related to the statistical analysis this should be clearly stated in the legend.

Response: Thanks for pointing out these unclear denotations. All statistical analyses are two-sided, unpaired t-test. Statistical significance was defined as not significant (ns), $P > 0.05$; * $P < 0.05$; ** $P < 0.01$; *** $P < 0.001$; **** $P < 0.0001$. Significance is indicated in figures. We clarified them in the revised manuscript.

Figure S3 – what is meant by “ABT263 tips binding pocket”?

Response: Sorry for the confusion. It has been corrected as ABT263 “chloride-benzene” moiety binding pocket.

Reviewer #2 (Remarks to the Author):

Review

The authors show the structure of the HBx protein BH3 domain bound to a Bcl domain that is substantially different from the structure determined by Jiang et al (PNAS (2016 113, 2074). They find the peptide site is shifted from the typical binding site and that the peptide they use does not form the typical all helical structure of other BH3 domains. They do not discuss if a longer peptide might give a different structure. They do observe that their peptide precipitates HBx and that a mutant peptide that specifically removes residues critical to the unusual binding site does not. Surprisingly they find their peptide, which can interact with Bcl but presumably not the SMC5/6 complex or the HBV genome, can activate HBV expression – if reproducible, this is a stunning observation. At a later point they show a truncated peptide inhibits HBV replication (I question whether this is more like and HBx- phenotype). They also test the importance of the unique BH3-Bcl interface they constructed a mutant HBx where two residues W120 and L123 of are mutated to A. The mutant did not work as well in cells or in mice, but there are few controls to test the basis of this attenuation, so the result is correlative not mechanistic. In a nice bit of chemical biology the authors show that their peptide and two ABT small molecules do not compete for the same Bcl site.

In summary, the authors show a HBx Bcl interaction that is different from the previous structure but do not consider that the rest of the HBx protein may also affect this structure.

They show how a long version of their peptide is HBV stimulatory while a short one appears to be inhibitory.

They show that two Bcl-specific small molecules bind via a non-overlapping site and do not interfere with the activity of HBV suppressive 118-127 peptide.

Response: Reviewer #2 commented that we reported some important and “stunning observations”, but raised some questions and concerns, which we have addressed in the revised manuscript and will discuss below.

Major points

The description of Bcl-xL is inadequate (p4 and p17). In the expressed protein, is residue 26 connected to 83? Where does 153 residues figure in (p17)? A protein composed of residues 1-26 and 83-197 is 141 residues long. Is the N-terminal Sumo-6His tag removed before crystallization? If so how? Was it visible in the crystal structure?

Response: We apologize for the confusion. Yes, we have screened many Bcl-xL protein expression vectors reported in the literature. The best expression construct is described in the experimental section as follows: “The codon optimized DNAs encoding Bcl-xL Δ TM (the flexible loop, residues 27–82, and the transmembrane domain, residues, 197–233, are deleted, and the resulting protein has 153 amino acids and was named as Bcl-xL thereafter) was ordered from Genewiz (www.genewiz.com). The vector expressing the recombinant Bcl-xL protein with an N-terminal SUMO tag and a 6 × His tag was generated by inserting the Bcl-xL PCR product into pET-SUMO vector (Invitrogen; www.thermofisher.com) and was verified by sequencing. The sequence of SUMO-6xHis-Bcl-xL is attached here.

MGSSHHHHHSSGLVPRGSHMASMSDSEVNQEAKPEVKPEVKPETHINLKVSDGSSEI
FFKIKKTTPLRRLMEAFKRQKEMDSLRFYDGIQADQTPEDLDMEDNDIIEAHREQI
GGSMSQSNRELVDFLSYKLSQKGYWSQMAAVKQALREAGDEFELRYRRAFSDLTS
QLHITPGTAYQSFEQVVELFRDGVNWRIVAFFSFGGALCVESVDKEMQVLVSRIAAW
MATYLNDHLEPWIQENGGWDTFVELYGNNAEAESRKGQER, with the sequence of Bcl-xL highlighted by underline. The SUMO tag was removed by SUMO Protease and the Bcl-xL protein further purified before crystallization trials. We included more detail in the Methods section in the revised manuscript as suggested.

The description of the BH3-domain binding pocket is inadequate. Top of page 5, the text refers to the movement of $\alpha 3$ relative to $\alpha 4$, but I have to go to supplementary info to identify the helices. Insert a new panel into figure 1. Is the pocket for the HBX peptide smaller? By what measure? Considering the extended C-terminus, is the buried surface area that different? The data in figure 1f do not show the differences in the residues bound. The lower panels of figure 1 do not appear to be in the same orientation. The electrostatic surface appears to be the standard Pymol vacuum surface, but is not identified as such. If the Bim-peptide structure has a different version of Bcl-xL, especially with regard to differences in the beginning and end points of visible peptide, it will change the surface shading, but the change is not meaningful. Clarify the sequence overlap, trim the models the same, and give them the same orientation. How can there be a hydrogen bond from the side chain of Leu194 to Asp130 in HBx – the Leu side chain is aliphatic.

Response: Many thanks for the insightful comments. In Figure 1, we want to highlight the overall structural comparison of Bcl-xL bound to two different BH3 peptides. Yes, we agree that if we insert a new figure panel in Figure 1 showing the structural movement of $\alpha 3$ relative to $\alpha 4$, it would be better to clarify the movement of the helix. However, due to the space limit of Figure 1, the inclusion of a detailed structural figure could cause confusions for the readers to catch the main point that the binding of different BH3 motif peptide could trigger the overall structural changes both in Bcl-xL and in the bound peptide. Indeed, this is another good example of showing the “induced-fit” effect as observed in many structures.

We have used exactly the same orientations to make the figures shown at the top panel (Fig. 1a, b) and at the bottom panel (Fig. 1d, e). Yes, we used the standard pymol software and standard vacuum surface to make these figures and have indicated this in the figure legend. We would like to highlight that although the Bcl-xL construct used is almost identical to the one used in the literature, the two Bcl-xL structures have significant structural variations by comparing these two Bcl-xL structures bound to two different peptides, most likely due to the “induced-fit” effect. The structural differences observed in Figure 1 are true structural changes induced by different peptides, instead of a shading problem. We agree that the shading could be different in these two figure panels, which is mainly due to the significant structural deviations instead of sequence differences of these two Bcl-xL structures. Nevertheless, the main purpose of Figure 1 is to highlight the

significant differences in the peptide binding pocket before and after peptide binding.

We also apologize for the mistake that we made to describe the hydrogen bond. The right description should be “Moreover, a hydrogen bond is formed between the main chain of Leu194 in Bcl-xL and the side chain of Asp130 in the HBx BH3-like motif...”

Nevertheless, based on the reviewer’s suggestion, we remade Figs. 5a-d. We have trimmed the N-terminal residues of Bcls and superimposed all the structures based on secondary structures. The PDBID is labeled under each figure.

For figure 2, is it known that the mutant Bcl-xL is properly folded?

Response: Regarding the expression level and solubility, the mutant Bcl-xL behaves just like the wild-type Bcl-xL protein (please see Fig. 2B, lysate immunoblotting). In general, Alanine substitutions should have limited impact on protein folding. Alanine scanning is a very popular method to determine the contribution of a specific residue to the function of a given protein.

Figure 3. Why does delivery of a BH3-like peptide enhance production of virus markers to 2x background? Does this indicate that HBx interaction with Bcl is virus inhibitory? This possibility needs to be discussed.

Response: Sorry about the confusion. The HBV replicon used in this experiment is a HBx-deficient construct (pHBV1.3-Xnull) and has a greatly reduced ability in promoting replication and viral gene expression in HepG2 cells, as compared with wild-type HBV replicon. The HBx-aa113-135 peptide most likely interacts well with Bcl proteins like a full-length HBx protein and thus can substitute for the missing HBx protein to enhance the virus production, which indeed is the case as we have shown in Fig. 3f-i using full-length HBx protein.

Literature is not well cited. The references and PDB accession numbers for all other structures used should be in the text and in the figure legends.

Response: We thank the reviewer for this suggestion. The references have been cited and the PDB accession numbers have been added in the revised text and in the revised figure legends.

I would like to see electron density for the whole BH3 peptide. The use of electrostatic surfaces without adequate labeling and with different protein constructs is confusing. The backbone trace in Fig S1 is cleaner.

Response: We thank the reviewer for this suggestion. A new supplementary Figure 1 (Fig. S1) showing the simulated anneal omit map covering the whole BH3 peptide has been added in this revised version. The map is displayed with the sigma value contoured at 2.0.

Minor points

Pymol uses a non-standard designation of cartoon for ribbon diagrams. It is better to describe ribbon diagrams as such.

Figure 1. panels c, d, e, and f are incorrectly identified

Response: Many thanks for the helpful comments. We agree that our figures may not be the perfect ones. However, we indeed used the default values to make the figures presented in Fig. 1. We also used the similar setting to make the figures for many other published manuscripts.

Figure 4. two panels are labeled 'e'. figure organization is difficult

Response: Thanks for the comment. We corrected these in revised manuscript.

Figure 5. The panels are too small and should be made larger. The panels show ribbon diagram of peptide BH3-like domains or stick figure diagrams of drugs bound to

electrostatic surfaces of the Bcl2 domain. The authors have a more confusing description of the figures. In panel a, the Bcl domain is in a different orientation than in the other panels, making comparison difficult.

Response: We thank the reviewer for the comments. We agree that the figures could be a little difficult to compare, probably due to the significant structural differences. However, we would like to point out that the Bcl-xL construct and the peptide we used are different from ones published in literature. Moreover, the significant conformational/structural changes induced by peptide binding make the exact structural superimposition (same orientation) impossible. Hence, we use the DALI server to superimpose the structures first to make the similar orientation as close as we can. We are sorry that we are not able to make the exact same orientation of two structures with significant conformational/secondary structure deviations.

Figure 6. Labels a and b are not present

Response: We added the labels in revised manuscript.

Reviewer #3 (Remarks to the Author):

In this manuscript, Zhang et al described structural and functional analyses of binding between HBV X protein and Bcl family of proteins and defined the interacting domains and residues for the two proteins. This interaction was uncovered a few years ago based on the presence of a putative BH3-like domain on HBx and later supported by some functional studies. The structural biology analysis is done well and rigorously, and appears to be convincing in this manuscript. This reviewer, however, questions the premise that HBx indeed possesses a true BH3-like domain. The original homology alignment (Geng, et al, PNAS 2012; Jiang et al, PNAS 2016) between HBx and authentic BH3-domain-containing proteins, like BAX, BAD, etc was at best marginal. The current structural analyses actually indicated that the HBX BH3-like domain binds to a different region from that of the well-known BH3 binding region on Bcl family of proteins. Furthermore, the HBx BH3-like domain does not exert a similar functional effect as that of

the known BH3-mimetics, like ABT-263 and ABT-737. Thus, based on structural and functional criteria, this reviewer is not convinced that that HBx indeed contains a BH3-like domain.

On the other hand, this region of HBx is obviously important to the function of HBx by mutagenesis analysis, which had been shown previously by many other reports on the functions of HBx. While a HBx peptide containing this region can bind to Bcl-xL physically in a highly artificial condition, as shown in this paper, it does not mean that such an interaction indeed occurs in a cellular environment. The fact that another peptide with 3 additional amino acids binds to a different region on Bcl-2, raises concern regarding the physiological nature of this putative binding domain. The key question of what is the true biological target of this region of HBx remains unresolved.

In this and previous studies, the authors have performed some functional studies by using HBV replication system to support the interaction between HBx and Bcl-2 family of proteins. The studies were performed in transfection system, which is prone to artefacts. HBV infectious system, such as HepG2-NTCP, is readily available now and should be used to validate the functional relevance of this interaction. It is well known that knocking down Bcl family of proteins can induce apoptosis and other untoward effects, and possibly causes nonspecific and indirect perturbation of HBV replication. Thus, findings based on knocking down Bcl-2 need to be interpreted with caution.

Response:

We thank the reviewer for the thoughtful comments and suggestions. In order to confirm the physiological significance of the HBx/Bcl-xL interaction, we validate the importance of the HBx interface residues in HBV replication in HepG2-NTCP cells that supporting HBV infection as suggested by the reviewer. In literature, HepG2-NTCP cell line is believed to be a more physiological model for HBV replication and viral protein expression. To test this, we produced different recombinant HBV particles, HBV-WT, HBV-WL/AA, and HBV-Xnull, respectively, in Huh7 cells by transfection with pHBV1.3, pHBV1.3-Xnull, and pHBV1.3-WL/AA replicons. These HBV particles were normalized to the same viral titer to infect HepG2-NTCP cells. Consistent with the in vivo data from the hydrodynamically

injected HBV mouse model, the levels of HBsAg and HBeAg in the culture medium of HepG2-NTCP cells infected by HBV-WL/AA particles were much lower than those from HepG2-NTCP cells infected by HBV-WT particles, while viral antigen levels were even under lower limit of detection (Fig. 4 i, j).

We also assessed the HBV inhibitory activity of the HBx118-127 peptide in HepG2-NTCP cells. A serial of different concentrations of HBx118-127 peptide were used to treat the HepG2-NTCP cells during HBV infection, significant reduction of HBeAg and HBV DNA levels were achieved at a dose-response manner (Fig. 6e,f), while there was no obvious cytotoxicity at the concentrations used (Fig. 6g).

These new results obtained from the HepG2-NTCP cells provide additional strong support to our conclusion that the HBx/Bcl-xL interaction is physiologically important for HBV replication and viral gene expression.

It is also puzzling that a HBx mimetic on one hand, can restore the replication defect of a HBV mutant without HBx, but on the other hand, inhibits WT HBV replication.

Response: This is a very good question. Our results show that the longer HBx BH3-like peptide can restore the replication defect of a HBV mutant replicon lacking functional HBx, whereas the short HBx mimetic inhibits WT HBV replication. We speculate that the HBx-aa113-135 peptide likely binds to Bcl-xL in a way that closely resembles the interaction between full-length HBx and Bcl-xL and thus can complement the loss of HBx to promote HBV replication. In contrast, the shorter peptide HBx-aa118-127, albeit being able to occupy the binding pocket of Bcl-xL and blocking the binding of full-length HBx to Bcl-xL, lacks additional N-terminal (aa113-117) and C-terminal (aa128-135) residues that may be crucial for the long peptide to mimic the functional interaction of HBx with Bcl-xL. In this sense, the shorter peptide HBx-aa118-127 behaves as a dominant-negative form of HBx and the longer peptide HBx-aa113-135 acts as a partial mimetic for HBx. Determination of the structure of the full-length HBx/Bcl-xL complex should shed light on this very interesting question.

Minor comments:

1. The legend in Fig. 1 was mislabeled with regard to the panels.

Response: Thanks for pointing out this. We have corrected the figure legend in the revised manuscript.

2. The discussion should also provide a more balanced view of the field with respect to other important recent findings regarding the functions of Hbx in HBV replication.

Response: We thank the reviewer for this suggestion. In the discussion section of the revised manuscript, we added a brief paragraph on HBx and its reported roles on HBV replication as follow: "Studies based on HepaRG cells³⁰ and human-liver-chimeric mice³¹ have indicated that HBx is required to initiate and maintain HBV replication. The complex interactions of HBx with multiple host factors, such as DDB1^{32, 33}, SMC5/6³⁴, and Bcl2^{6, 9}, and their important roles in HBV replication, are described in many studies and reviews.^{30, 35, 36} .."

3. The concept of using a HBx mimetic to block HBV replication, as shown in Fig. 7 is not convincing. The inhibitory effects on various HBV replication markers, are not very impressive. Should control for viability and effect on unrelated expression constructs.

Response:

We agree that the inhibitory effects of the HBx-aa-118-127 peptide mimetic, albeit significant, are not very impressive. That is why we indicated that this peptide mimetic is a promising drug lead. However, with further refinements and modifications, we may obtain much more potent HBV inhibitors, much the same way as the evolvement of the BH3-only mimetics, which took many refinement steps.

In this revised manuscript, we also provided crucial data showing significant reduction of HBeAg and HBV DNA levels at a dose-response manner when the HBx-aa118-127 peptides were used to treat the HepG2-NTCP cells during HBV infection (Fig. 6e,f). Moreover, as suggested by the reviewer, we assessed the cytotoxicity of the

HBx-aa118-127 peptide treatment. CCK8 assay was used and we found that it has no obvious cytotoxicity at concentrations between 12.5-200 $\mu\text{g}/\text{mL}$ (Fig. 6g). Taken together, our data demonstrate that a HBx BH3-like mimetic can indeed block HBV replication in cell models.

Reviewers' Comments:

Reviewer #1:

Remarks to the Author:

The authors have addressed all my concerns adequately. The additional ITC data is compelling, and supports their notion that the binding sites for ABT737 and Hbx peptide are largely non-overlapping.

The authors have addressed a number of the issues raised by reviewer 2, however several items are not adequately addressed.

The authors must provide Circular dichroism data for wild type and mutant Bcl-xL in order to demonstrate that mutant proteins were folded. Simply stating that Ala substitutions do not tend to unfold proteins is insufficient, and IPs do not adequately demonstrate correct folding.

The authors must provide the total buried surface area for the different complexes (their own, Bcl-2-hbx, complexes with ABT737 and 263). This should be supported by a superimposition of the ABT737/263 complexes with their own complex (as trace for the Bcl-2 backbone to ensure – in pymol this is referred to as ribbon setting), with the ligands (hbx or abt737/263 as sticks).

Reviewer #4:

Remarks to the Author:

The study provides accurate structural and functional analyses of interactions that can emerge between HBx protein and the cellular Bcl proteins. Although this is not the first report pointing out interactions between HBx and the bcl family of proteins, the analyses performed here appear solid. It is plausible that if such interactions, if occurring also in HBV chronically infected livers, may play a crucial role in HBV-mediated pathogenesis. Particularly intriguing is the observation that an HBx peptide can “rescue” replication of an HBx-defective mutant, while a shorter HBx peptide appears to be inhibitory. The latter observation is quite important since it may have some potential for the development of future anti HBV agents. Both hydrodynamic injection experiments and infection experiments performed in HepG2-NTCP cells provide clear evidence regarding the ability of the described HBx mutants to affect HBV replication levels.

However, the mechanisms by which HBx could impact HBV replication through the binding with bcl-xL remain unclear and should be further discussed. Moreover, the authors should consider the possibility that the HBx mutants and peptides here used may also affect interactions with other host factors.

In line with the comments made by reviewer #3, I was puzzled by the fact that the authors pay little attention to the consequences that the HBx mutants here described may have regarding the smc5/6 complex, in particular when the peptides are used. I think some considerations should be done in this direction to strengthen the meaning of the findings here reported and to put them in the context of other main changes that HBx is known to induce on cellular proteins. All in all, I am not questioning the data regarding the interactions between HBx and bcl, but I am missing data supporting a direct role for this interaction in modulating HBV replication, whereas the effects of the same HBx constructs on smc5/6 remain completely uninvestigated.

Specific comments.

1) I perfectly agree with the comments made by reviewer #3 regarding the point that “The discussion should also provide a more balanced view of the field with respect to other important recent findings

regarding the functions of HBx in HBV replication". Although this was improved in the revised version, I still find awkward that one of the major breakthrough made regarding HBx function (the capacity to induce degradation of the smc5/6 complex in vitro and in vivo) is not even mentioned in the introduction. The present statement: "The HBV genome encodes four viral gene products, one of which, the HBV X protein (HBx), plays an important role in stimulating viral gene expression and replication by altering host gene expression patterns, blocking immune response, and elevating cytosolic calcium 3-6" is misleading and needs to be changed to give proper weight to more relevant recent findings (e.g. Decorsiere, Nature 2016).

2) As mentioned in the general comments, the authors should address the questions whether the mutants used in this study and that were shown to impair HBV activity, also affect the interaction of HBx with DDB1, thus leading to impaired degradation of the smc complex. Since HBx was clearly shown to induce degradation of the smc5/6 complex, it would be important to assess whether the HBx-aa113-135 peptide, which the authors show to be able to restore HBV replication, is also sufficient to promote smc5/6 degradation. On the same line, it would be important to determine whether the HBV-WL/AA mutations – which were shown here to strongly affect HBV replication – also impair DDB1 binding or at least are associated with reduced degradation of the smc5/6 complex. For this, staining of the smc6 protein in HepG2-NTCP cells treated with the different HBV (HBx) constructs would be sufficient.

Below is our point-by-point response to the comments of the reviewers

Reviewer #1:

The authors have addressed all my concerns adequately. The additional ITC data is compelling, and supports their notion that the binding sites for ABT737 and Hbx peptide are largely non-overlapping.

The authors have addressed a number of the issues raised by reviewer 2, however several items are not adequately addressed.

Response: We thank Reviewer #1 for the positive summary and comments of our study and findings and our efforts to address all his/her concerns.

The authors must provide Circular dichroism data for wild type and mutant Bcl-xL in order to demonstrate that mutant proteins were folded. Simply stating that Ala substitutions do not tend to unfold proteins is insufficient, and IPs do not adequately demonstrate correct folding.

Response: In the revised manuscript, we have provided circular dichroism (CD) data for wild type and mutant Bcl-xL proteins (Fig. S7). The fitting CD curves of wild type and mutant Bcl-xL proteins indicate that the Ala substitutions (L52, E73 and D77) do not change the overall structure of Bcl-xL. Please see the figure below:

Fig. S7. Circular dichroism data for wild type and mutant Bcl-xL proteins. (a) A SDS-PAGE image of wild type and mutant recombinant Bcl-xL proteins as indicated. (b) Circular dichroism spectra of wild type and mutant Bcl-xL proteins.

We included the CD data of wild type and mutants Bcl-xL proteins in Fig. S7, but we did not comment on these data in the revised manuscript, because we did not use any of these mutant Bcl-xL proteins in this study.

The authors must provide the total buried surface area for the different complexes (their own,

Bcl-2-hbx, complexes with ABT737 and 263). This should be supported by a superimposition of the ABT737/263 complexes with their own complex (as trace for the Bcl-2 backbone to ensure – in pymol this is referred to as ribbon setting), with the ligands (hbx or abt737/263 as sticks).

Response: We thank the reviewer for this good suggestion. In the revised manuscript, we provided the structural superimposition of the Bcl-xL/HBx-BH3-like complex, Bcl-xL/ABT-263 complex, Bcl-xL/ABT-737 complex, and Bcl-2/HBx-BH3-like complex (Fig. 5e, f). The total buried surface area for each of these complexes is provided in the Table S2. The total buried surface area of the Bcl-xL/HBx-BH3-like complex is 2,014.76 Å² per molecule, suggesting that the Bcl-xL/HBx-BH3-like complex is physiologically relevant.

Reviewer #4:

The study provides accurate structural and functional analyses of interactions that can emerge between HBx protein and the cellular Bcl proteins. Although this is not the first report pointing out interactions between HBx and the bcl family of proteins, the analyses performed here appear solid. It is plausible that if such interactions, if occurring also in HBV chronically infected livers, may play a crucial role in HBV-mediated pathogenesis. Particularly intriguing is the observation that an HBx peptide can “rescue” replication of an HBx-defective mutant, while a shorter HBx peptide appears to be inhibitory. The latter observation is quite important since it may have some potential for the development of future anti HBV agents. Both hydrodynamic injection experiments and infection experiments performed in HepG2-NTCP cells provide clear evidence regarding the ability of the described HBx mutants to affect HBV replication levels.

Response: We thank Reviewer #4 for the positive summary and comments of our study and findings.

However, the mechanisms by which HBx could impact HBV replication through the binding with bcl-xL remain unclear and should be further discussed. Moreover, the authors should consider the possibility that the HBx mutants and peptides here used may also affect interactions with other

host factors.

In line with the comments made by reviewer #3, I was puzzled by the fact that the authors pay little attention to the consequences that the HBx mutants here described may have regarding the smc5/6 complex, in particular when the peptides are used. I think some considerations should be done in this direction to strengthen the meaning of the findings here reported and to put them in the context of other main changes that HBx is known to induce on cellular proteins. All in all, I am not questioning the data regarding the interactions between HBx and bcl, but I am missing data supporting a direct role for this interaction in modulating HBV replication, whereas the effects of the same HBx constructs on smc5/6 remain completely uninvestigated.

Response: We agree that the HBx mutations used in this study may also affect HBx interactions with other host factors and have performed additional experiments to examine if the HBx mutations used in this study affect HBx binding to DDB1 and affect the Smc6 protein levels as suggested by the reviewer. Our data show that the W120A/L123A double mutations in the HBx BH3-like domain do not affect the interaction between HBx and DDB1 in HepG2 cells (Fig. S8a) and do not impair the function of HBx in promoting degradation of SMC6 (Fig. S8b, c). We also provided the data indicating that the HBx-aa113-135 peptide does not interact with DDB1 and has no effect on the SMC6 protein level (Fig. S9).

Two of our previous studies have provided strong evidence and a model regarding how interactions of HBx with Bcl-2 and Bcl-xL proteins could impact HBV replication (Geng et al, PNAS 109, 18465-18470, 2012; Geng et al, PNAS 109, 18471-18476, 2012). In these two studies, interactions of HBx with Bcl-2 and Bcl-xL proteins have been shown to result in elevation of cytosolic calcium, which is critical for HBV viral replication. Fig. 5 from one of the papers (Geng et al, PNAS 109, 18471-18476, 2012) summarizes the key results and the model and is shown below. In particular, RNAi knockdown of Bcl-xL or Bcl-2 reduces HBx-induced cytosolic Ca²⁺ elevation (Fig. S1 of the same paper, not shown here) and HBV replication. The importance of elevated cytoplasmic calcium to HBV

replication, transcription, and other aspects of the viral life cycle has been reported by several other studies (Choi Y et al. *Virology* 332: 454–463, 2005; Bouchard et al. *Science* 294:2376–2378, 2001; Bouchard et al. *J Virol* 77:7713–7719, 2003; Oh et al. *Exp Mol Med* 35:301–309, 2003; Lara-Pezzi et al. *EMBO J* 17:7066–7077, 1998). As suggested by the reviewer, we briefly discuss the importance and the underlying mechanism of HBx interactions with Bcl-2 and Bcl-xL to HBV replication in both Introduction and Discussion.

Fig. 5. Bcl-2 and Bcl-xL are important for HBV DNA replication. (A and B) (Right) HepG2 cells infected by lentivirus expressing control, Bcl-2, or Bcl-xL shRNA were transfected with the pHBV replicon and subjected to Q-PCR analysis as described in Fig. 3D. (Left) One portion of the cells was analyzed by immunoblotting to examine the expression levels of Bcl-2 and Bcl-xL, using α -tubulin as a loading control. Data are presented as mean \pm SEM. *** $P < 0.0001$. (C) A working model of HBx-dependent viral pathogenesis. **HBx directly interacts with Bcl-2 and Bcl-xL to increase cytosolic Ca^{2+} .** Increased cytosolic Ca^{2+} then promotes HBV replication and cell death.

Specific comments.

1) I perfectly agree with the comments made by reviewer #3 regarding the point that “The discussion should also provide a more balanced view of the field with respect to other important recent findings regarding the functions of HBx in HBV replication”. Although this was improved in the revised version, I still find awkward that one of the major breakthrough made regarding HBx function (the capacity to induce degradation of the smc5/6 complex in vitro and in vivo) is not even mentioned in the introduction. The present statement: “The HBV genome encodes four viral gene products, one of which, the HBV X protein (HBx), plays an important role in

stimulating viral gene expression and replication by altering host gene expression patterns, blocking immune response, and elevating cytosolic calcium 3-6” is misleading and needs to be changed to give proper weight to more relevant recent findings (e.g. Decorsiere, Nature 2016).

Response: We thank the reviewer for the comments and suggestion. In the revised manuscript, we substantially modified the introduction to reflect this recent important finding as follow: “The HBV genome encodes four viral gene products, one of which, the HBV X protein (HBx), plays an important role in HBV life cycle through interaction with a number of host proteins^{3-5, 6, 7, 8}. For example, HBx has been shown to interact with Damage Specific DNA Binding Protein 1 (DDB1), which could redirect the DDB1-containing E3 ubiquitin ligase to target the structural maintenance of chromosome 5/6 complex (Smc5/6) for degradation, unleashing the transcriptional repression by Smc5/6 to increase HBV viral gene expression^{6, 9}.”

We also discussed this important finding again in the Discussion section: “The interactions of HBx with multiple host factors, such as DDB1^{3, 4}, SMC5/6⁶, Bcl-2 and Bcl-xL^{7, 8}, and their important roles in HBV replication and viral gene expression, have been reported in a number of studies and reviews.^{5, 31, 33}..... As a multi-functional viral protein, HBx has been shown to interact with a number of host factors^{3-5, 6, 7, 8}, including DDB1, Bcl-2 and Bcl-xL. The W120A/L123A mutations in the HBx BH3-like motif could potentially affect HBx interactions with other host factors. One with particular interest is the DDB1 protein, a component of the DDB1-CUL4-ROC1 E3 ligase complex, that interacts with HBx to target Smc5/6 for ubiquitylation and degradation^{6, 9}, which relieves inhibition of HBV virus gene expression by Smc5/6”

2) As mentioned in the general comments, the authors should address the questions whether the mutants used in this study and that were shown to impair HBV activity, also affect the interaction of HBx with DDB1, thus leading to impaired degradation of the smc complex. Since HBx was clearly shown to induce degradation of the smc5/6 complex, it would be important to assess whether the HBx-aa113-135 peptide, which the authors show to be able to restore HBV replication,

is also sufficient to promote smc5/6 degradation. On the same line, it would be important to determine whether the HBV-WL/AA mutations – which were shown here to strongly affect HBV replication – also impair DDB1 binding or at least are associated with reduced degradation of the smc5/6 complex. For this, staining of the smc6 protein in HepG2-NTCP cells treated with the different HBV (HBx) constructs would be sufficient.

Response:

We thank the reviewer for these helpful comments. To verify if the HBx mutations used in this study affect their interaction with DDB1 and thus degradation of the Smc complex, we performed two experiments. First, we used biotin-conjugated HBx-aa113-135 wild type and mutant peptides to pull down DDB1 by Co-IP assays. Our data showed that DDB1 could not be pulled down by the HBx-aa113-135 peptides (Fig. S9a), indicating that the HBx-aa113-135 peptide does not bind DDB1. To verify if the HBx-aa113-135 peptides affect the protein level of SMC6, we transfected the HepG2 cells with the wild-type and mutant HBx-aa113-135 peptides and detected the level of SMC6 by immunostaining. Our data showed that the levels of the SMC6 protein in transfected cells were comparable in HepG2 cells transfected with the HBx-aa113-135-WT, HBx-aa113-135-WL/AA, and control peptides (Fig. S9b, c). These results suggest that the HBx-aa113-135 peptide does not promote HBV replication and transcription through the DDB1/SMC6 pathway. Second, we transfected plasmids expressing full-length HBx-WT and HBx-WL/AA into HepG2 cells and examined the interaction between HBx and DDB1 by co-IP assays 48 hours later using an anti-HBx antibody. Consistent with the published results, DDB1 could be pulled down by the wild-type HBx protein (Fig. S8a). However, comparable amounts of DDB1 were pulled down by the mutant HBx-WL/AA proteins (Fig. S8a), indicating that the W120A/L123A double mutations in HBx do not affect the interaction between HBx and DDB1. Furthermore, we detected similar levels of the SMC6 proteins by immunofluorescence staining in HepG2 cells transfected with pTT22m-HBx-WT and pTT22m-HBx-WL/AA, but significantly higher SMC6 protein expression in HepG2 cells transfected with pTT22m vector control (Fig. 8b, c). These results again indicate that the W120A/L123A double mutations in HBx do not compromise the function of HBx in

degrading the SMC complex. Of note, Li et al reported that HBx binds DDB1 through an α -helical motif, including HBx-aa88-100 (Nature Structural & Molecular Biology 17,105-111, 2009), which does not overlap with the HBx-aa113-135 BH3-like motif.

Fig. S8. The W120A/L123A mutations in the HBx BH3-like motif do not affect the interaction of HBx with DDB1 and do not impair the function of HBx on degradation of SMC6 in HepG2 cell. a, Full length HBx interacts with DDB1 in HepG2 cells. Coimmunoprecipitation (Co-IP) experiments were performed in HepG2 cells transfected with the pTT22m-HBx-WT and pTT22m-HBx-(W120A, L123A) plasmids, respectively. Cell lysate was incubated and then precipitated by an anti-HBx antibody (16F9) and analyzed by immunoblotting using an anti-DDB1 antibody (Abcam, ab109027) and anti-HBx antibody (16F9). b-c, HepG2 cells were transfected with the pTT22m-HBx-WT,

pTT22m-HBx-WL/AA, and pTT22m vector, respectively. 48 hours post transfection, transfected cells were stained with an anti-SMC6 antibody (Thermo Fisher Scientific, # PA5-45522) and an Alexa Fluor 488 Donkey Anti-Rabbit IgG (H+L) secondary antibody (Invitrogen, A-21206) to reveal the Smc6 protein expression and an anti-HBx mouse monoclonal antibody (16F9) and an Alexa Fluor 680 Donkey Anti-Mouse IgG secondary antibody (Invitrogen, A-10038) to reveal HBx protein expression. Nuclei were stained with DAPI (Invitrogen, D1306). The fluorescence was detected by Opera Phenix High Content Screening System (PerkinElmer) and the mean green fluorescence intensity of transfected cells (mCherry-positive) was quantified by Opera Phenix High Content Screening System. The data shown are mean \pm SD. Significant differences between groups are indicated. **** $P < 0.0001$; two-tailed unpaired t -tests.

Fig. S9. The HBx HBx-aa113-135 peptides do not interact with DDB1 and do not affect the level of the Smc6 proteins in HepG2 cell. a, Coprecipitation experiments were performed with HepG2 cell lysate that was incubated with biotinylated HBx-aa113-135 peptides (WT) and mutant peptides with the W120A/L123A mutations, respectively. The pull-down proteins were analyzed by immunoblotting (IB) using an anti-DDB1 antibody (Abcam, ab109027). b-c, HBx-aa113-135 peptides (wild-type and the W120A/L123A double mutant peptides) were delivered into HepG2 cells using PEP-1 nanoparticles, respectively. 48 hours post transfection, cells were stained with an anti-SMC6 antibody (Thermo Fisher Scientific, # PA5-45522) and an Alexa Fluor 488 Donkey Anti-Rabbit IgG (H+L) secondary antibody (Invitrogen, A-21206) to reveal the

Smc6 protein expression. Nuclei were stained with DAPI (Invitrogen, D1306). The fluorescence was detected by Opera Phenix High Content Screening System (PerkinElmer) and the mean green fluorescence intensity of cells was quantified by Opera Phenix High Content Screening System. The data shown are mean \pm SD. Significant differences between groups are indicated on the top. ns, no significant difference; two-tailed unpaired *t*-tests.

Reviewers' Comments:

Reviewer #1:

Remarks to the Author:

The authors have addressed all my concerns now. Marc Kvansakul

Reviewer #4:

Remarks to the Author:

The authors have addressed most of the concerns that were raised in the revision process. In particular, they modified both the introduction and the discussion as required. They also performed new experiments to assess the impact of the different HBx mutants and peptides on the degradation of the smc5/6 complex. Unfortunately, I found the data shown in Figure S8 and S9, and in particular the immunofluorescence, not convincing. Entirely in contrast with previous published data, the staining seems to be unspecific, since smc6 is not detected in the nucleus. Smc6 is known to co-localize with PML in nucleus, but not to give such a cytoplasmic or membrane like staining. In infected cells, there should be no positive smc6 staining. This would be a proper control that the authors should use. Thus, I would suggest that either the authors test different antibodies (as those used in previously published papers) and provide proper controls for the immunofluorescence staining, or they should remove this part of the results and simply state that further studies are needed to assess the impact of these peptides and HBx mutants on smc5/6 degradation.

The authors have addressed all my concerns now. Marc Kvensakul

We want to thank Dr. Marc Kvensakul for his very helpful comments and suggestions.

Reviewer #4 (Remarks to the Author):

The authors have addressed most of the concerns that were raised in the revision process. In particular, they modified both the introduction and the discussion as required. They also performed new experiments to assess the impact of the different HBx mutants and peptides on the degradation of the smc5/6 complex. Unfortunately, I found the data shown in Figure S8 and S9, and in particular the immunofluorescence, not convincing. Entirely in contrast with previous published data, the staining seems to be unspecific, since smc6 is not detected in the nucleus. Smc6 is known to co-localize with PML in nucleus, but not to give such a cytoplasmic or membrane like staining. In infected cells, there should be no positive smc6 staining. This would be a proper control that the authors should use.

Thus, I would suggest that either the authors test different antibodies (as those used in previously published papers) and provide proper controls for the immunofluorescence staining, or they should remove this part of the results and simply state that further studies are needed to assess the impact of these peptides and HBx mutants on smc5/6 degradation.

Response: Thank you very much for your comments and suggestions. We removed the results about smc5/6 and stated that further studies are needed to assess the impact of these peptides and HBx mutants on smc5/6 degradation.